# Phosphorylation of seryl-tRNA synthetase by ATM/ATR is essential for hypoxia-induced angiogenesis

Yi Shi[1,2☯]*, Ze Liu[1☯], Qian Zhang[1], Ingrid Vallee[1], Zhongying Mo[1], Shuji Kishi[3], Xiang-Lei Yang[1]*

**1** Department of Molecular Medicine, The Scripps Research Institute, La Jolla, California, United States of America, **2** School of Medicine, Nankai University, Tianjin, China, **3** Department of Metabolism and Aging, The Scripps Research Institute, Jupiter, Florida, United States of America

☯ These authors contributed equally to this work.
* yishi@nankai.edu.cn (YS); xlyang@scripps.edu (X-LY)

**Data Availability Statement:** All relevant experimental data are within the paper and its Supporting Information files.

## Abstract

Hypoxia-induced angiogenesis maintains tissue oxygen supply and protects against ischemia but also enhances tumor progression and malignancy. This is mediated through activation of transcription factors like hypoxia-inducible factor 1 (HIF-1) and c-Myc, yet the impact of hypoxia on negative regulators of angiogenesis is unknown. During vascular development, seryl-tRNA synthetase (SerRS) regulates angiogenesis through a novel mechanism by counteracting c-Myc and transcriptionally repressing vascular endothelial growth factor A (VEGFA) expression. Here, we reveal that the transcriptional repressor role of SerRS is inactivated under hypoxia through phosphorylation by ataxia telangiectasia mutated (ATM) and ataxia telangiectasia mutated and RAD3-related (ATR) at Ser101 and Ser241 to attenuate its DNA binding capacity. In zebrafish, SerRS$^{S101D/S241D}$, a phosphorylation-mimicry mutant, cannot suppress VEGFA expression to support normal vascular development. Moreover, expression of SerRS$^{S101A/S241A}$, a phosphorylation-deficient and constitutively active mutant, prevents hypoxia-induced binding of c-Myc and HIF-1 to the *VEGFA* promoter, and activation of VEGFA expression. Consistently, SerRS$^{S101A/S241A}$ strongly inhibits normal and tumor-derived angiogenesis in mice. Therefore, we reveal a key step regulating hypoxic angiogenesis and highlight the importance of nuclear SerRS in post-developmental angiogenesis regulation in addition to vascular development. The role of nuclear SerRS in inhibiting both c-Myc and HIF-1 may provide therapeutic opportunities to correct dysregulation of angiogenesis in pathological settings.

## Introduction

Oxygen is critical for the development and growth of most multicellular organisms. Sophisticated molecular mechanisms have been evolved to sense and respond to changes in oxygen levels in order to maintain cell and tissue homeostasis. Therefore, the hypoxia response is protective against ischemic tissue injury. On the other hand, pathophysiological hypoxia signaling

**Funding:** This work was supported by grants from the National Institutes of Health [R01 GM088278 and R01 NS113583] to X.-L.Y., the National Natural Science Foundation of China [81772974] to Y.S., aTyr Pharmacy (https://www.atyrpharma. com) through an agreement with Scripps Research, and a fellowship from the National Foundation for Cancer Research (https://www.nfcr. org) to Z. L. The funders had no role in study design, data collection and analysis, decision to publish or preparation of the manuscript.

**Competing interests:** I have read the journal's policy and the authors of this manuscript have the following competing interests: X.-L.Y. is a founder of aTyr Pharma. The remaining authors declare no competing interests.

**Abbreviations:** 3′-UTR, 3′ untranslated region; ARNT, aryl hydrocarbon receptor nuclear translocator; ATM, ataxia telangiectasia mutated; ATR, ataxia telangiectasia mutated and RAD3-related; ChIP, chromatin immunoprecipitation; control-MO, control morpholino; EMSA, electrophoresis mobility shift assay; GlyRS, glycyl-tRNA synthetase; GTP, guanosine 5′-triphosphate; HIF-1, hypoxia-inducible factor 1; HRE, hypoxia-response element; HUVEC, human umbilical vein endothelial cell; IACUC, Institutional Animal Care and Use Committee; IL-8, interleukin 8; ISV, intersegmental vessel; K-RAS (or KRAS), Kirsten rat sarcoma viral oncogene homolog; MO, morpholino; NF-κB, nuclear factor kappa B; PHD, prolyl hydroxylase; PI3K, phosphoinositide-3 kinase; PTU, 1-phenyl-2-thiourea; qRT-PCR, real-time quantitative reverse transcription PCR; RNAi, RNA interference; ROS, reactive oxygen species; SBS, SerRS binding site; SerRS, seryl-tRNA synthetase; SerRSWT, wild-type SerRS; sh-control, control shRNA; shRNA, short hairpin RNA; siRNA, small interfering RNA; TCA, trichloroacetic acid; VEGFA, vascular endothelial growth factor A; VHL, von Hippel–Lindau protein; WT, wild-type.

also contributes to the development of diseases, such as cardiovascular disorders and solid tumors [1–4].

One of the many aspects of the hypoxia response is the stimulation of angiogenesis through the production of angiogenic factors, especially vascular endothelial growth factor A (VEGFA) [5–7]. The response is mediated mainly through a transcription factor called hypoxia-inducible factor (HIF)-1 [8]. HIF-1 is a heterodimer consisting of an oxygen-regulated HIF-1α subunit and a constitutively expressed HIF-1β subunit, also known as aryl hydrocarbon receptor nuclear translocator (ARNT). In the presence of oxygen, the HIF-1α subunit is rapidly hydroxylated by prolyl hydroxylases (PHDs) and subsequently ubiquitylated through von Hippel–Lindau protein (VHL) and degraded by the 26S proteasome [9,10]. However, in the absence of oxygen, HIF-1α subunit is stabilized; the HIF-1 heterodimer is formed and, in turn, translocated to the nucleus to bind to the hypoxia-response element (HRE) in the promoter region of *VEGFA* to activate gene expression [6,8,10–12].

In addition to HIF-1, other transcription factors such as c-Myc have been reported to mediate hypoxia responses and stimulate VEGFA expression [12–16]. In fact, hypoxia-induced angiogenesis is preserved in HIF-1-deficient colon cancer xenograft [14,15,17]. In a recent clinical trial of EZN-2968, the antisense oligonucleotide inhibitor of HIF-1α successfully down-regulated both mRNA and protein levels of HIF-1α but failed to reduce the expression of VEGFA in tumor biopsies [18]. These preclinical and clinical observations highlight the significant role of HIF-1-independent regulation of VEGFA and angiogenesis in tumor progression.

In addition to the contribution of transcription factors that positively regulate angiogenesis, it is conceivable that a complete hypoxia response would require the collaboration between positive and negative transcriptional regulators. However, in contrast to the extensive understanding of how transcriptional activators respond to hypoxia, little is known on the response of negative regulators during hypoxia. We have recently identified seryl-tRNA synthetase (SerRS) as a key transcriptional repressor of VEGFA [19,20]. SerRS is well known as a member of the aminoacyl-tRNA synthetase family responsible for charging serine onto its cognate tRNA to generate substrates for protein biosynthesis in the cytosol. Surprisingly, forward genetic studies in zebrafish suggested a critical role of SerRS in vascular development independent of its aminoacylation activity [21–23]. Interestingly, this role of SerRS occurs in the nucleus, and a nucleus-targeting domain is found in all vertebrate SerRS [20]. Nuclear SerRS regulates the vasculature by transcriptionally repressing VEGFA expression and does so through binding to a 27-bp DNA on the *VEGFA* promoter and recruiting histone deacetylase SIRT2 for epigenetic gene silencing [19]. This role of SerRS directly counteracts that of c-Myc, the major transcriptional factor regulating VEGFA during development [19,24,25]. Because c-Myc is involved in hypoxia-stimulated VEGFA production [16], we postulated that the anti-c-Myc activity of SerRS would need to be attenuated in order to achieve VEGFA stimulation under hypoxia.

Through a series of studies in human cells, zebrafish, and mice, we found that this is indeed the case. The transcriptional repressor role of SerRS is inactivated during hypoxia through phosphorylation of SerRS by ataxia telangiectasia mutated (ATM) and the related kinase ATR to attenuate its DNA-binding capacity. Importantly, both c-Myc and HIF-1 cannot bind to the *VEGFA* promoter without SerRS being phosphorylated and removed from the DNA. Therefore, our findings reveal a missing piece in the regulation of angiogenesis during hypoxia and demonstrate the importance of inactivating negative regulators in this process. Such understanding may provide new opportunities in treating ischemic cardiovascular diseases and cancer.

## Results

### SerRS inhibition of *VEGFA* transcription is blocked under hypoxia

To investigate whether SerRS is involved in the hypoxia response to regulate VEGFA expression, we knocked down SerRS in HEK293 cells with a short hairpin RNA (shRNA) targeting the 3′ untranslated region (3′UTR) of the SerRS gene. Under normal levels of oxygen (normoxia), as we observed previously [19], VEGFA expression is up-regulated upon knocking down SerRS, compared with control cells transfected with a nonspecific control shRNA (sh-control) or an shRNA targeting a different aminoacyl-tRNA synthetases (i.e., glycyl-tRNA synthetase (GlyRS); sh-GlyRS) (Fig 1A and 1B, S1A Fig). Under hypoxia, VEGFA expression is significantly increased in control cells as expected (Fig 1A and 1B). However, the stimulation is largely lost in SerRS knockdown cells (Fig 1A and 1B), suggesting under hypoxia VEGFA level is no longer dependent on SerRS. We hypothesize that this observation indicates a hypoxia-mediated inactivation of the angiostatic function of SerRS.

### SerRS is phosphorylated at serine 101 and serine 241 by ATM/ATR during hypoxia

To test the hypothesis, we first examined the effect of hypoxia on the expression of SerRS and did not observe an obvious change (S1B Fig). We then investigated the possibility that SerRS is inactivated under hypoxia through posttranslational modifications. In a large-scale mass spectrometry study, SerRS was found to be phosphorylated at serine 241 (S241) by ATM/ATR kinases, which are activated under DNA damage [26]. In PhosphoSitePlus database [27], we found another possible SerRS phosphorylation site serine 101 (S101). Both sites have the conserved ATM/ATR substrate motif with a serine or threonine followed by a glutamine and preceded by 2 hydrophobic residues (at −1 and −3 positions relative to the serine/threonine) (Fig 1C) [28]. Multiple sequence alignment revealed strict conservation of S/T101 and S/T241 and flanking ATM/ATR substrate motif residues in vertebrate SerRS (Fig 1C), concurrent with the emergence of the role of SerRS in regulating vascular development and angiogenesis [20].

DNA fragment-induced SerRS phosphorylation was confirmed in vitro by $^{32}$P-labeling (Fig 1D). Double-stranded DNA oligonucleotides were added to the nuclear extract of HEK293 cells to mimic DNA damage in order to activate ATM/ATR [29]. The "activated" nuclear extract specifically induced robust phosphorylation of the purified recombinant SerRS but not of GlyRS (Fig 1D). SerRS phosphorylation was further verified by using specific phosphor-ATM/ATR substrate (p-S*Q) antibody (Fig 1E). To confirm the phosphorylation sites on SerRS, we substituted S101 and S241 with alanine separately (S101A, S241A) and simultaneously (SerRS$^{S101A/S241A}$ or in abbreviation SerRS$^{AA}$). SerRS$^{S101A}$ showed a decreased level of phosphorylation, whereas SerRS$^{S241A}$ and SerRS$^{AA}$ had little phosphorylation in vitro as revealed by both p-SQ antibody detection (Fig 1E) and $^{32}$P-labeling (S1C Fig), suggesting that SerRS can be phosphorylated by ATM/ATR kinases at both S101 and S241 and that S241 is the major phosphorylation site on SerRS.

Next, we investigated whether SerRS is phosphorylated in hypoxic conditions, under which ATM/ATR is known to be activated [30–32]. Indeed, phosphorylation of endogenous SerRS was detected in HEK293 and human umbilical vein endothelial (HUVEC) cells under hypoxia within 12 h (Fig 1F and 1G). Interestingly, there is an apparent temporal delay between the activation of ATM/ATR (approximately 3 h after hypoxia exposure) and the phosphorylation of SerRS in HEK293 cells (at least 6 h after hypoxia exposure) (Fig 1F). However, this delay was not observed in HUVEC cells, and phosphorylation of SerRS was detected 3 h after hypoxia exposure (Fig 1G). Possibly, an unknown inhibitor exists in HEK293 cells, but not in

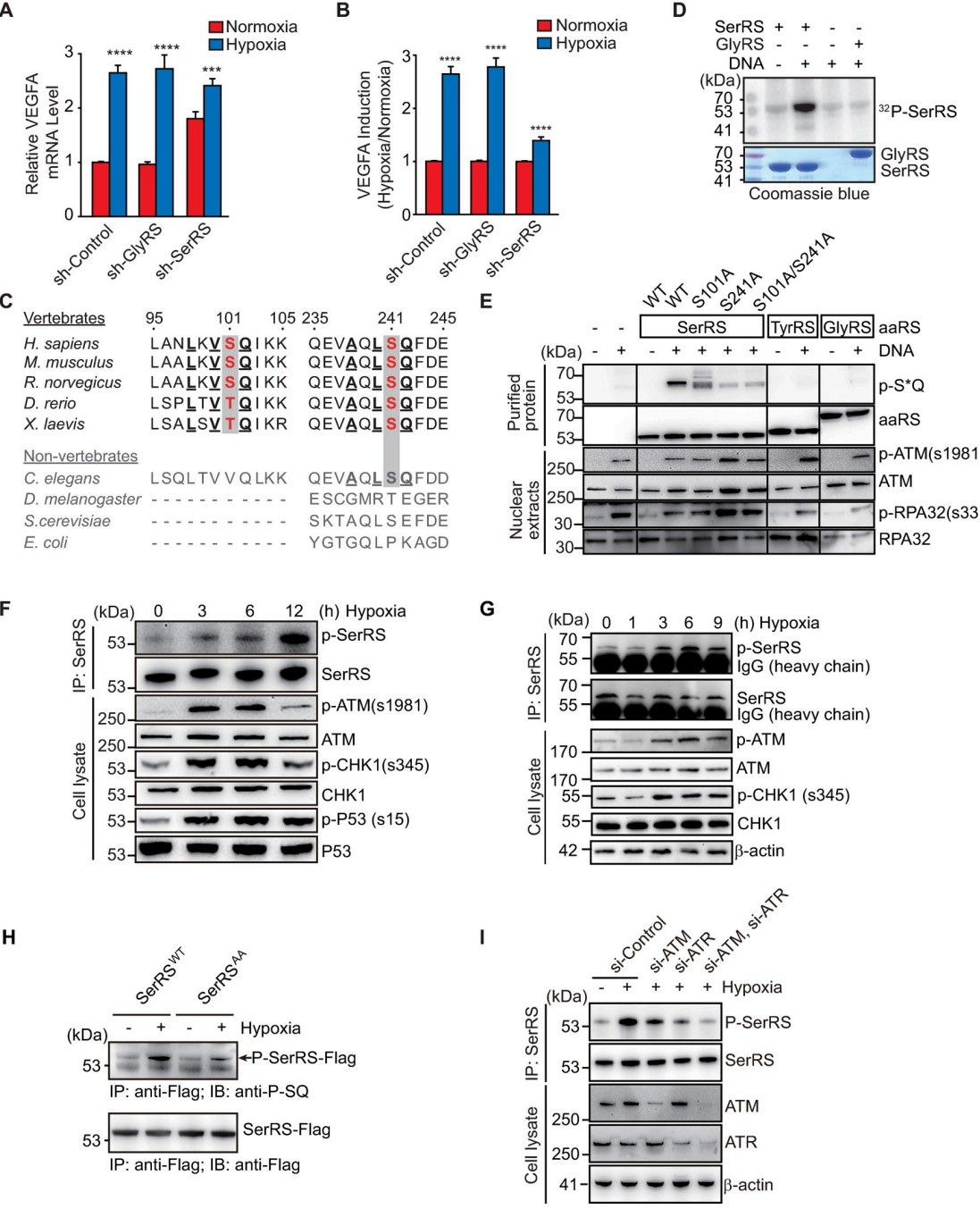

**Fig 1. SerRS is involved in the hypoxia response to induce VEGFA expression through phosphorylation by ATM and ATR kinases at S101 and S241 residues.** (**A**, **B**) qRT-PCR analysis of VEGFA expression in HEK293 cells transfected with sh-SerRS, sh-GlyRS, or nonspecific sh-Control under hypoxia or normoxia conditions. VEGFA levels (**A**) and relative induction of VEGFA under hypoxia (**B**) were plotted as means ± SEM ($n = 4$, biological replicates, Student $t$ test, $^{**}p < 0.01$, $^{****}p < 0.0001$). (**C**) Sequence alignment of SerRS proteins from vertebrates and non-vertebrates flanking serine 101 and serine 241 (highlighted in red) sites numbered according to the human sequence. The conserved ATM/ATR substrate motif residues are underlined. (**D**) $^{32}$P-labeling to confirm phosphorylation of SerRS, but not GlyRS, in vitro using recombinant tRNA synthetases. Double-stranded DNA oligonucleotides are used to mimic DNA damage to activate ATM/ATR from the nuclear extract of HEK293 cells. (**E**) Western blot analysis to confirm recombinant SerRS, but not GlyRS and TyrRS, is phosphorylated by ATM/ATR by using a specific antibody against phosphor-ATM/ATR substrate (p-S*Q). The purified His$_6$-tagged aaRS proteins are recognized by anti-His$_6$-tag antibody. Activation of ATM and ATR was confirmed by autophosphorylation of ATM (at S1981) and phosphorylation of RPA32 (at S33), respectively. (**F**, **G**) IP and western blot analysis to confirm hypoxia-induced SerRS phosphorylation in HEK293 (**F**) and HUVEC (**G**) cells. IP was performed with anti-SerRS antibody from mouse and rabbit for HEK293 (**F**) and HUVEC (**G**)

cells, respectively. Phosphorylated SerRS was detected with the antibody against phosphor-ATM/ATR substrate (p-S*Q) from rabbit. Activation of ATM and ATR was confirmed by autophosphorylation of ATM (at S1981) and phosphorylation of CHK1 (at S345) and P53 (at S15). (**H**) Hypoxia-induced phosphorylation of SerRS in HEK293 cells is reduced in SerRS[AA] compared with SerRS[WT]. (**I**) Hypoxia-induced phosphorylation of SerRS was decreased when ATM and ATR were knocked down separately or together by siRNAs (si-ATM, si-ATR). At least 2 biological replicates were performed for western blot experiments with consistent results. Representative images were shown. See S1 Data for quantitative data and statistical analysis. See S2 Data for original, uncropped images supporting blots and gel results. aaRS, aminoacyl-tRNA synthetase; ATM, ataxia telangiectasia mutated; ATR, ataxia telangiectasia mutated and RAD3-related; GlyRS, glycyl-tRNA synthetase; HUVEC, human umbilical vein endothelial cell; IP, immunoprecipitation; qRT-PCR, real-time quantitative reverse transcription PCR; SerRS, seryl-tRNA synthetase; SerRS[AA], S101A/S241A; SerRS[WT], wild-type SerRS; sh-Control, control shRNA; sh-GlyRS, shRNAs targeting GlyRS; sh-SerRS, shRNAs targeting SerRS; siRNA, small interfering RNA; TyrRS, tyrosyl-tRNA synthetase; VEGFA, vascular endothelial growth factor A.

HUVEC cells, that needs to be removed prior to the phosphorylation of SerRS by ATM/ATR. Interestingly, SerRS[AA], when exogenously expressed in hypoxic HEK293 cells, was phosphorylated to a lesser extent than the exogenously expressed wild-type SerRS (SerRS[WT]) (Fig 1H), confirming that S241 and/or S101 are the phosphorylation sites.

To further confirm that ATM and ATR are responsible for SerRS phosphorylation under hypoxia, we knocked down ATM and ATR either separately or simultaneously by RNA interference. Hypoxia-induced phosphorylation of SerRS was greatly inhibited when either ATM or ATR was knocked down, and completely blocked when both kinases were knocked down simultaneously (Fig 1I). Consistent with these results, SerRS phosphorylation under hypoxia could also be blocked by the specific ATM inhibitor KU-55933 [33] and ATR inhibitor VE-821 [34] (S1D Fig).

It is worth noting that we used a 2-enzyme approach to induce hypoxia in our cell-based experiments [35]. Glucose oxidase to deplete oxygen, whereas catalase was used to convert $H_2O_2$ (produced by the prior oxidation reaction) to water to prevent DNA damage in cells. Because ATM and ATR are well known to be activated by DNA damage, we measured H2AX phosphorylation (γ-H2AX) to ensure lack of DNA damage during the 2-enzyme treatment (S2 Fig). Indeed, no obvious increase in gamma-H2AX level was observed in hypoxic cells, confirming that the activation of ATM/ATR is not an artifact of the 2-enzyme treatment.

## Phosphorylation inactivates SerRS regulating VEGFA in cells and in vivo

To understand if phosphorylation of SerRS affects its role as a transcriptional repressor, we created a mutant form of SerRS bearing double substitutions of S101 and S241 with aspartic acid residues (SerRS[S101D/S241D] or in abbreviation SerRS[DD]) to mimic the phosphorylated SerRS. In contrast to SerRS[WT] and SerRS[AA], SerRS[DD] can no longer repress *VEGFA* transcription in HEK293 cells (Fig 2A), suggesting that phosphorylation of SerRS inhibits its transcriptional repressor activity.

To investigate the effect of SerRS phosphorylation in vivo, we took advantage of a previously well-established zebrafish system [19,20,22]. In zebrafish embryos, endogenous SerRS was knocked down by an antisense morpholino (SerRS-MO), which led to a 4-fold increase in the mRNA level of *vegfaa* (Fig 2B). This effect can be completely and partially rescued by co-injection of human SerRS[WT] mRNA and SerRS[AA] mRNA with SerRS-MO, respectively. In fact, SerRS[WT] and SerRS[AA] should be functionally equivalent in this experiment because the zebrafish are not under hypoxia; the partial activity of SerRS[AA] is likely due to the lower expression level of SerRS[AA] compared with SerRS[WT] (Fig 2B). However, co-injection of SerRS[DD] mRNA, which was expressed at a similar level as SerRS[AA], had no rescue effect (Fig 2B), confirming that phosphorylations at S101 and S241 block the transcriptional repressor activity of SerRS on *VEGFA* in vivo.

We further examined the effect of SerRS phosphorylation on vascular development in zebrafish. Knocking down SerRS by injecting SerRS-MO into the yolk of zebrafish embryos at 1-

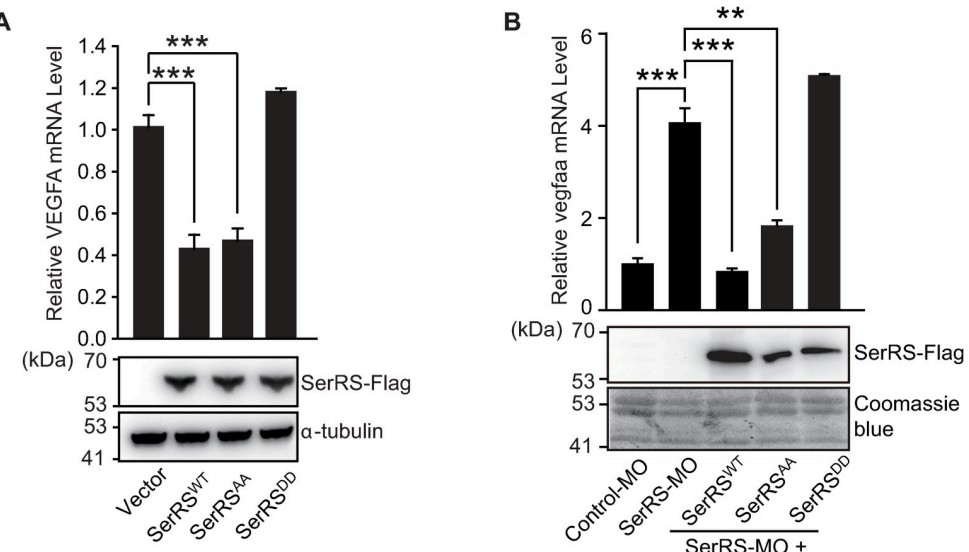

**Fig 2. SerRS phosphorylation at S101 and S241 inhibits its function in repressing VEGFA expression in cellulo and in vivo.** (**A**) qRT-PCR analysis of VEGFA expression in HEK293 cells transfected with Flag-tagged SerRS$^{WT}$ or mutants with SerRS$^{AA}$ and SerRS$^{DD}$ substitutions. The exogenous expressions of the SerRS proteins were confirmed by immunoblotting against the Flag tag. VEGFA expression levels are plotted as means ± SEM ($n = 3$, biological replicates, $^{****}p < 0.0001$, Student $t$ test). (**B**) qRT-PCR analysis of *vegfaa* transcription in zebrafish injected with SerRS-MO and with co-injection of SerRS-MO and SerRS$^{WT}$, SerRS$^{AA}$, or SerRS$^{DD}$ mRNA. Data plotted as means ± SEM ($n = 3–4$, biological replicates, $^{**}p < 0.01$, $^{***}p < 0.001$, Student $t$ test). The Coomassie blue staining was used to show equal total protein loading. See S1 Data for quantitative data and statistical analysis. See S2 Data for original, uncropped images supporting blots and gel results. qRT-PCR, real-time quantitative reverse transcription PCR; SerRS, seryl-tRNA synthetase; SerRS$^{AA}$, S101A/S241A; SerRS$^{DD}$, S101D/S241D; SerRS-MO, antisense morpholino against SerRS; SerRS$^{WT}$, wild-type SerRS; VEGFA, vascular endothelial growth factor A.

to 2-cell stage resulted in abnormal hyper-intersegmental vessel (ISV) branching phenotype in 70% ($n = 147$ out of 211) of zebrafish larvae at 72 h postfertilization (hpf) (S3A and S3B Fig) as expected [19,20,22]. In contrast, only 9% ($n = 13$ out of 142) of zebrafish embryos injected with a control morpholino (control-MO) exhibited the hyper-ISV phenotype. Co-injection of human SerRS$^{AA}$ mRNA largely rescued the abnormal ISV branching (26%, $n = 33$ out of 125), which is comparable to the rescue effect of SerRS$^{WT}$ mRNA (18%, $n = 29$ out of 162) (S3A and S3B Fig). In contrast, SerRS$^{DD}$ could not rescue the abnormal ISV branching (63%, $n = 84$ out of 134) (S3A and S3B Fig), confirming that SerRS phosphorylation blocks its antiangiogenic activity in vivo.

## Phosphorylation inactivates SerRS by attenuating its DNA binding capacity

To explore the molecular mechanism of how SerRS phosphorylation inactivates its function as a transcriptional repressor, we first examined the effect of hypoxia on SerRS nuclear localization in HEK293 cells. Hypoxia does not affect the nuclear localization of SerRS (S4A Fig). Consistently, we also found similar cytoplasmic versus nuclear distributions of exogenously expressed SerRS$^{WT}$, SerRS$^{DD}$, and SerRS$^{AA}$ proteins in HEK293 cells (S4B Fig).

Given that SIRT2 is a necessary cofactor for SerRS to epigenetically silence VEGFA expression [19], we next examined the impact of hypoxia on the interaction between SerRS and SIRT2. Exposure to hypoxia did not obviously affect the amount of SIRT2 co-immunoprecipitated with SerRS in HEK293 cells (S4C Fig). Consistently, SIRT2 interacts with SerRS$^{AA}$ and SerRS$^{DD}$ as strongly as with SerRS$^{WT}$ (S4D Fig), indicating a lack of effect of hypoxia on the SerRS-SIRT2 interaction.

Finally, we explored the interaction of SerRS with the *VEGFA* promoter. As detected by electrophoresis mobility shift assay (EMSA), the direct binding between recombinant SerRS protein and a $^{32}$P-labeled 27-bp DNA fragment previously identified as the SerRS binding site from the *VEGFA* promoter[19] was strongly attenuated with the phosphor-mimicking mutant SerRS$^{DD}$ but not the phosphor-deficient mutant SerRS$^{AA}$ (Fig 3A). Importantly, this decreased DNA binding of SerRS$^{DD}$ was not caused by mutation-induced protein misfolding, because both SerRS$^{DD}$ and SerRS$^{AA}$ proteins have largely maintained the enzymatic activity in tRNA aminoacylation (S5 Fig).

Consistently, in HEK293 cells, ectopically expressed SerRS$^{DD}$ mutant also showed a decreased binding to *VEGFA* promoter as determined by a chromatin immunoprecipitation assay (Fig 3B). Thus, we concluded that hypoxia-induced phosphorylation blocks the transcriptional repressor activity of SerRS by weakening its DNA binding capacity. Interestingly, the S101 and S241 phosphorylation sites are located near the proposed DNA binding sites of SerRS [19] (Fig 3C). Phosphorylation of these serine residues would introduce negative charges to create electrostatic repulsion between SerRS and DNA. This could explain why phosphorylation of SerRS weakens its DNA binding capacity.

## SerRS phosphorylation is required for hypoxia-induced c-Myc and HIF-1 binding to the *VEGFA* promoter

In previous work, we have established that SerRS and c-Myc compete with each other for binding to the *VEGFA* promoter [19]. Therefore, as SerRS is phosphorylated and loses its DNA-binding capacity, we would expect an enhanced binding of c-Myc to the *VEGFA* promoter. Indeed, during hypoxia, the amount of endogenous SerRS bound to the *VEGFA* promoter in both HUVEC and HEK293 cells gradually decreases, concurrent with a gradual increase in the level of c-Myc bound to the *VEGFA* promoter (Fig 3D and 3E). Remarkably, the amount of HIF-1α bound to the *VEGFA* promoter also increases in a trajectory parallel to that of c-Myc (Fig 3D and 3E), raising the possibility that the removal of SerRS from the *VEGFA* promoter is also necessary for HIF-1 to bind to the DNA. It is worth noting that the release of SerRS from the VEGFA promoter starts earlier in HUVEC than in HEK293 cells (Fig 3D and 3E), which is consistent with the distinct time course of SerRS phosphorylation in these 2 cell types under hypoxia (Fig 1F and 1G). Nevertheless, the binding of HIF and Myc to the *VEGFA* promoter happens around the same time, i.e., 6 h after hypoxia exposure, in both HEK293 and HUVEC cells (Fig 3D and 3E).

To confirm that SerRS phosphorylation is essential for hypoxia-induced binding of c-Myc and HIF-1α to the *VEGFA* promoter, we compared DNA binding of c-Myc and HIF-1α in HEK293 cells that express SerRS$^{WT}$ versus phosphor-deficient SerRS$^{AA}$ and phosphor-mimicry SerRS$^{DD}$. As expected, while the binding of SerRS$^{WT}$ to the *VEGFA* promoter decreases under hypoxia, binding of SerRS$^{AA}$ and SerRS$^{DD}$ no longer respond to change in the oxygen level (Fig 3F). The amount of c-Myc and of HIF-1α bound to the *VEGFA* promoter significantly increases during hypoxia in control cells expressing SerRS$^{WT}$ (Fig 3F). However, the increase is blocked when SerRS$^{AA}$ is expressed (Fig 3F). These results strongly indicate that the SerRS phosphorylation event is a prerequisite for hypoxia-induced binding of c-Myc and HIF-1 to *VEGFA*. Consistently, hypoxia-induced VEGFA expression is inhibited in HEK293 cells expressing SerRS$^{AA}$ but not SerRS$^{WT}$ (Fig 3G).

Interestingly, we found no difference in HIF-1α binding to the *VEGFA* promoter between cells expressing SerRS$^{WT}$ and SerRS$^{DD}$, whereas SerRS$^{DD}$ expression, compared with SerRS$^{WT}$, stimulates c-Myc binding to the *VEGFA* promoter with or without hypoxia (Fig 3F). This is likely due to the fact that c-Myc is constitutively present in the cell, whereas HIF-1α protein is

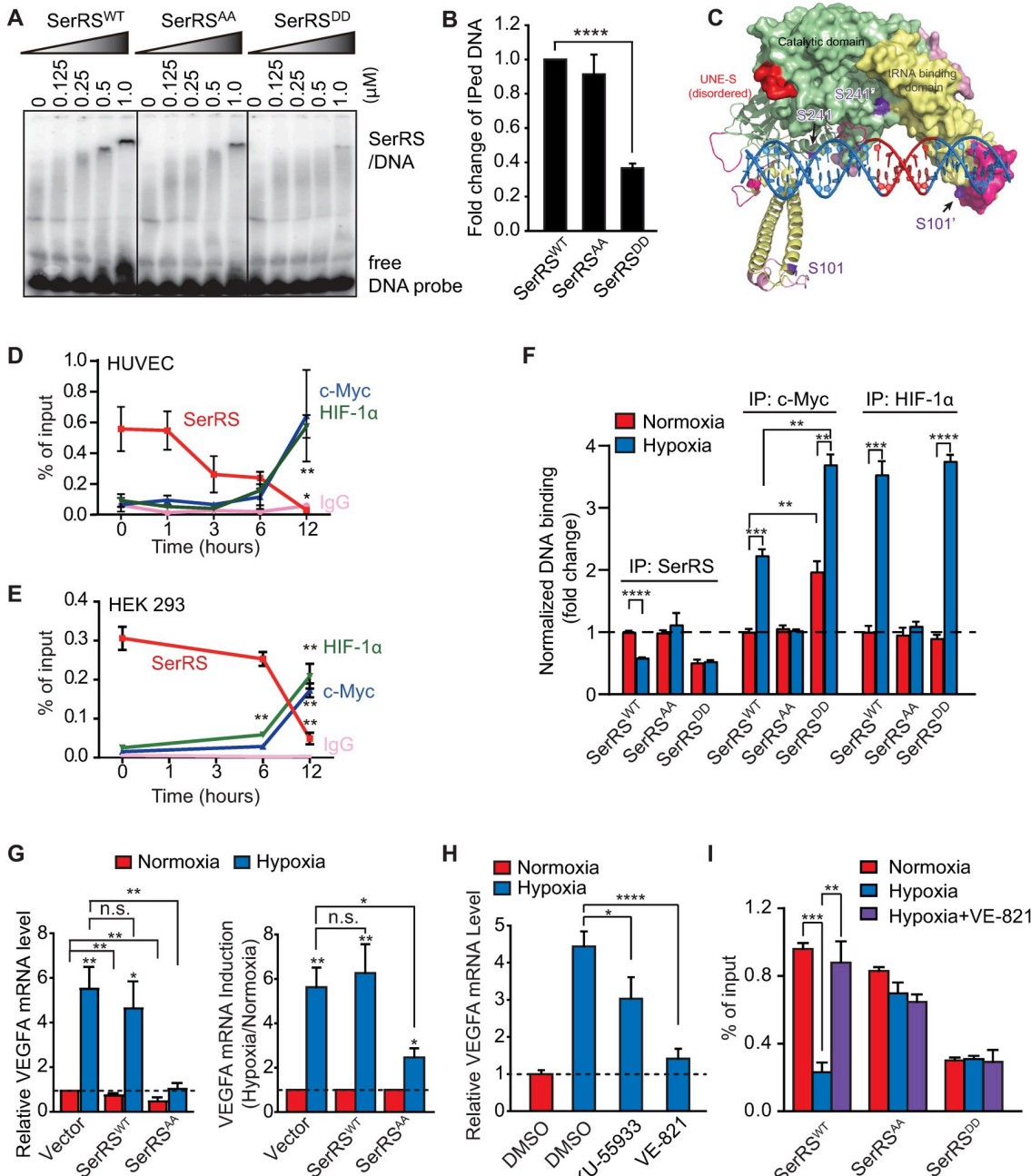

**Fig 3. ATM/ATR-caused SerRS phosphorylation attenuates its binding to *VEGFA* promoter, which allows c-Myc and HIF-1 to activate *VEGFA* transcription.** (**A**) EMSA to measure the binding of SerRS[WT], SerRS[AA], or SerRS[DD] with $^{32}$P-labeled DNA fragments (27 bp) corresponding to the SerRS binding site on human *VEGFA* promoter. (**B**) Chromatin-immunoprecipitated DNA by SerRS antibody was quantified by qPCR (ChIP-qPCR) to measure the binding of SerRS[WT], SerRS[AA], or SerRS[DD] on *VEGFA* promoter in HEK293 cells. The data were calculated as percentage of input DNA and then normalized against the SerRS[WT] group (means ± SEM, $n = 2$, biological replicates, ****$p < 0.0001$, Student $t$ test). (**C**) Structure model of SerRS dimer binding to *VEGFA* promoter DNA showing that the phosphorylation sites S101 and S241 are located near the predicted SerRS-DNA binding sites. (**D**, **E**) ChIP-qPCR to follow the binding of endogenous SerRS, c-Myc, and HIF-1α on *VEGFA* promoter during hypoxia in HUVEC cells (**D**) and HEK293 cells (**E**) (means ± SEM, $n = 3$, biological replicates, **$p < 0.01$, Student $t$ test). (**F**) ChIP-qPCR to measure the binding of SerRS, c-Myc, and HIF-1α on *VEGFA* promoter in HEK293 cells stably transfected with SerRS[WT], SerRS[AA], or SerRS[DD] (tag-free) under normoxia and hypoxia (12 h) conditions (means ± SEM, $n = 3$, biological replicates, **$p < 0.01$, ***$p < 0.001$, ****$p < 0.0001$, Student $t$ test). The expression level of the exogenous SerRS proteins (WT, AA, and DD) is similar (as shown in Fig 4A and S5 Fig). (**G**) qRT-PCR to measure VEGFA expression in HEK293 cells transfected with SerRS[WT] or SerRS[AA] under normoxia and hypoxia conditions (means ± SEM, $n = 4$, biological replicates, *$p < 0.05$, **$p < 0.01$, n.s., Student $t$ test). (**H**) qRT-PCR to

measure the effects of specific ATM inhibitor KU-55933 and specific ATR inhibitor VE-821 on hypoxia-induced VEGFA expression in HEK293 cells (means ± SEM, $n = 2$, biological replicates, $^*p < 0.05$, $^{****}p < 0.0001$, Student $t$ test). (**I**) ChIP-qPCR to measure the binding of SerRS to *VEGFA* promoter in HEK293 cells stably transfected with SerRS[WT], SerRS[AA], or SerRS[DD] under normoxia or hypoxia, in the presence or absence of VE-821 (means ± SEM, $n = 3$, biological replicates, $^{**}p < 0.01$, $^{***}p < 0.001$, Student $t$ test). See S1 Data for quantitative data and statistical analysis. See S2 Data for original, uncropped images supporting blots and gel results. ATM, ataxia telangiectasia mutated; ATR, ataxia telangiectasia mutated and RAD3-related; ChIP-qPCR, chromatin immunoprecipitation–quantitative real-time PCR; EMSA, electrophoresis mobility shift assay; HIF-1; hypoxia-inducible factor 1; HUVEC, human umbilical vein endothelial cell; n.s., not significant; qRT-PCR, real-time quantitative reverse transcription PCR; SerRS, seryl-tRNA synthetase; SerRS[AA], S101A/S241A; SerRS[DD], S101D/S241D; SerRS[WT], wild-type SerRS; VEGFA, vascular endothelial growth factor A.

only stabilized during hypoxia (S6 Fig). Under normoxia, the binding of c-Myc to the *VEGFA* promoter can be inhibited by SerRS[WT] but not SerRS[DD]. In contrast, no HIF-1 is present in cell under normoxia to be inhibited by SerRS[WT].

We ruled out the possibility that SerRS directly interact c-Myc and HIF-1 to influence their DNA binding (S6 Fig). To further demonstrate that the transcriptional repression activity of SerRS is dependent on binding to the *VEGFA* promotor, we designed a luciferase assay. Our previous work has identified a SerRS binding site (SBS) on the VEGFA promotor to be a 27-bp region (−62 to −36) that overlaps with the E-box binding site of Myc (−49 to −44) [19] (S7A Fig). The HIF-binding site on the *VEGFA* promotor is located upstream (−975 to −968) [5]. We added the *VEGFA* promoter (−1,005 to +379) in front of a luciferase reporter gene to test the transcriptional activity of SerRS[AA]. Indeed, expression of SerRS[AA] inhibits the hypoxia-induced luciferase activity, and this inhibition is dependent on SerRS-DNA binding, as removal of SBS abolishes the inhibitory effect of SerRS[AA] (S7B Fig).

## ATM/ATR-SerRS is a key pathway that regulates hypoxia-induced VEGFA expression

The fact that expressing the phosphorylation-deficient SerRS[AA] to bypass the ATM/ATR regulation can significantly inhibit hypoxia-induced VEGFA expression (Fig 3G) suggests that the ATM/ATR-SerRS pathway is critical for cellular induction of VEGFA during hypoxia. To confirm the importance of this pathway, we knocked down ATM or ATR or both in HEK293 cells (S8A Fig). Either ATM or ATR knockdown inhibited VEGFA induction under hypoxia, and it is unclear whether the dual knockdown has a stronger effect (S8B Fig), suggesting that both ATR and ATM are major players in stimulating VEGFA expression during hypoxia. We also blocked ATM and ATR individually in HEK293 cells by specific inhibitors. ATR inhibitor VE-821 dramatically inhibited VEGFA induction under hypoxia, whereas the effect of ATM inhibitor KU-55933 was reduced but still statistically significant (Fig 3H). Consistently, ATR inhibitor VE-821 could restore the binding of SerRS to VEGFA promoter whereas it had no effect on SerRS[AA] and SerRS[DD] (Fig 3I), strongly suggesting that hypoxia activated ATR plays a key role in promoting the release of SerRS from *VEGFA* promoter by direct phosphorylation of SerRS at S101 and S241.

## SerRS phosphorylation is required for activating VEGFA-mediated tube formation under hypoxia

To understand if the effect of SerRS on VEGFA transcription can be manifested at the protein level and because VEGFA mainly exists as an extracellular protein, we detected VEGFA in the culture media of both HEK293 and HUVEC cells by ELISA. As shown in Fig 4A and 4B, expression of SerRS[WT] and SerRS[AA], but not the phosphor-mimicking SerRS[DD], reduced the level of VEGFA protein in both cell types cultured in normoxia, although the observed effect is

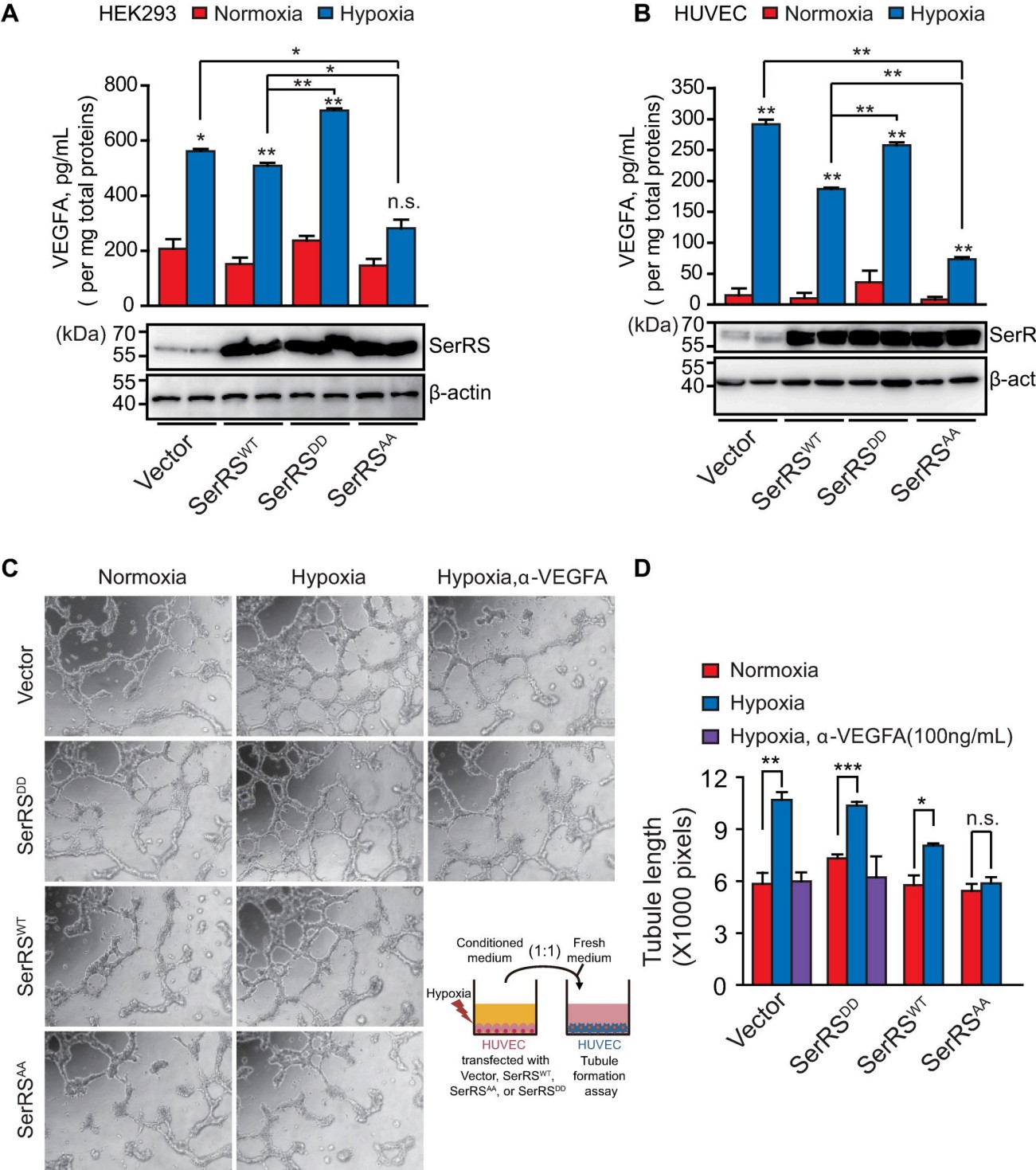

**Fig 4. SerRS phosphorylation is required for activating VEGFA-mediated tube formation under hypoxia.** (**A**, **B**) VEGFA levels in HEK293 (**A**) and HUVEC (**B**) cells stably transfected with SerRS$^{WT}$, SerRS$^{AA}$, or SerRS$^{DD}$ under normoxia or hypoxia conditions were measured by ELISA and normalized to total proteins in cell lysates. (means ± SEM, $n$ = 2 biological replicates, $^{*}p < 0.05$, $^{**}p < 0.05$, n.s., Student $t$ test). The SerRS protein levels were shown by western blot below the bar graph. (**C**, **D**) Endothelial cell tube formation assay (**C**) and quantification (**D**) using HUVEC cells treated with indicated conditional medium from wild-type or mutant SerRS stably transfected donor HUVEC cells cultured under normoxia or hypoxia conditions. The cartoon describes the experimental setup. The total length of tubules was measured using Image J software (means ± SEM, $n$ = 3, biological replicates, $^{*}p < 0.05$, $^{**}p < 0.01$, $^{***}p < 0.001$, n.s., Student $t$ test). See S1 Data for quantitative data and statistical analysis. See S2 Data for original, uncropped images supporting blots and gel results. HUVEC, human umbilical vein endothelial cell; n.s., not significant; SerRS, seryl-tRNA synthetase; SerRS$^{AA}$, S101A/S241A; SerRS$^{DD}$, S101D/S241D; SerRS$^{WT}$, wild-type SerRS; VEGFA, vascular endothelial growth factor A.

less obvious than that at the mRNA level (Fig 2A). Based on the ELISA analysis, the repressor activity of SerRS$^{WT}$ and SerRS$^{AA}$ on VEGFA protein level is similar at normal oxygen environment (Fig 4A and 4B). However, under hypoxia condition, the repressor activity of SerRS$^{AA}$ is significantly higher than that of SerRS$^{WT}$ (Fig 4A and 4B), in line with that SerRS$^{WT}$ but not SerRS$^{AA}$ can be phosphorylated and inactivated under hypoxia (Fig 1E and 1H, Fig 3G and 3I).

To further demonstrate the functional consequence of SerRS phosphorylation, we performed an endothelial cell tube formation assay with HUVEC cells cultured in different conditional media, which were obtained from culturing donor HUVEC cells stably transfected with various SerRS constructs (or an empty vector as the control) under normoxia or hypoxia. In the experiment where control donor cells (stably transfected with the empty vector) were involved, the conditional media from the hypoxia treatment group have much stronger activity in promoting HUVEC cell tube formation than the conditional media from the normoxia group, and the stimulation effect can be blocked by a VEGFA-neutralizing antibody, suggesting that the enhanced tube formation is mediated through hypoxia-induced VEGFA production from donor cells (Fig 4C and 4D). As expected, a similar result was found with the culture media from donor cells stably expressing the phosphor-mimicking SerRS$^{DD}$, because SerRS$^{DD}$ is no longer active in suppressing VEGFA. However, the stimulation effect is compromised with the culture medium of donor cells stably expressing SerRS$^{WT}$ and abolished with that of donor cells stably expressing the phosphorylation-deficient SerRS$^{AA}$ (Fig 4C and 4D). These cell-based experiments demonstrate that phosphorylation and inactivation of SerRS is critical for stimulating angiogenesis under hypoxia.

## SerRS$^{AA}$ bypasses hypoxia response and strongly inhibits angiogenesis in mice

Because ectopic expression of SerRS$^{AA}$ can block, to a large extent, the hypoxia-induced VEGFA expression in both HEK293 and HUVEC cells (Fig 3G and Fig 4A and 4B) and endothelial cell tube formation (Fig 4C and 4D), we next explored the activity of SerRS$^{AA}$ and the effect of SerRS phosphorylation in hypoxia-induced angiogenesis in mammals by using a matrigel plug angiogenesis assay. Mouse endothelial 3B11 cells were stably transfected with mouse SerRS$^{WT}$, SerRS$^{AA}$, or SerRS$^{DD}$ constructs to reach an expression level similar to that of the endogenous mouse SerRS (Fig 5A). The engineered 3B11 cells were mixed with matrigel in vitro at low temperature. The mixtures were injected subcutaneously into the mice and solidified into plugs, where a hypoxic environment would form prior to the induction of the vasculature. Two weeks after the injection, we confirmed, by the elevated HIF-1α protein level, that a hypoxia environment occurred inside the matrigel plugs (S9A Fig). At the same time, we evaluated the microvasculature in the plugs by VEGFA (Fig 5B and 5C) and CD31 (S9B and S9C Fig) immunostaining. As expected, expression of the phosphor-mimicking SerRS$^{DD}$ did not suppress microvascular formation, whereas the phosphor-deficient SerRS$^{AA}$ strongly suppress it, demonstrating that SerRS phosphorylation and inactivation is essential for hypoxia-induced angiogenesis in vivo. Importantly, the antiangiogenic activity of SerRS$^{WT}$ lies between that of SerRS$^{AA}$ and SerRS$^{DD}$, indicating that SerRS$^{WT}$ was partially inactivated under hypoxia by phosphorylation (Fig 5B and 5C).

## SerRS$^{AA}$ strongly inhibits tumor angiogenesis and tumor growth in mice

Hypoxia is a common phenomenon in solid tumors and leads to advanced and abnormal vascularization [2,3]. To evaluate whether hypoxia-induced SerRS phosphorylation and inactivation is also important for tumor angiogenesis and tumor growth, we stably transfected human triple negative breast cancer MDA-MB-231 cells with constructs expressing human SerRS$^{WT}$,

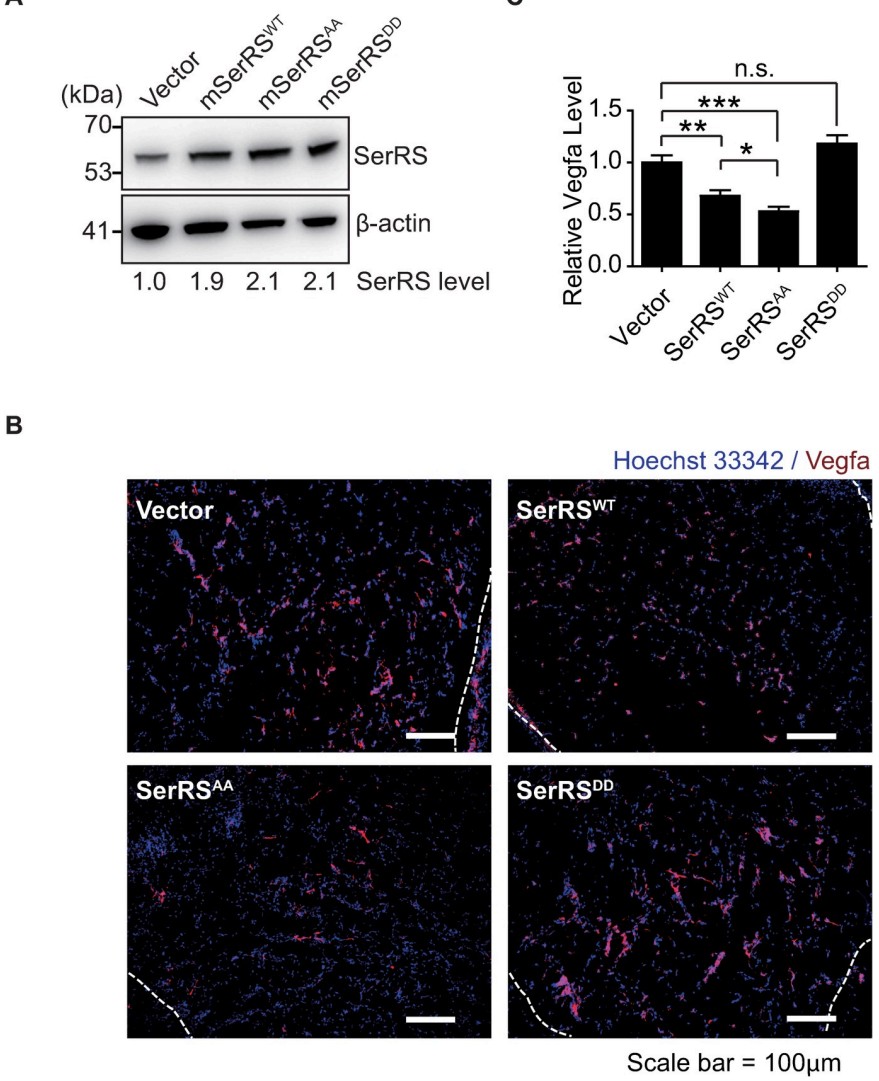

**Fig 5. SerRS^AA exhibits stronger activity than SerRS^WT in suppressing angiogenesis in mice.** (A) Western blot to measure the expression of SerRS proteins in mouse 3B11 endothelial cells stably transfected with tag-free mouse SerRS^WT or mutants (SerRS^AA and SerRS^DD). The SerRS levels are quantified by the density of the bands and indicated relative to the level of the endogenous SerRS (Vector). (B, C) Matrigel plug angiogenesis assay performed with stably transfected 3B11 cells in C3H/HeJ mice. Matrigel plugs (dash lines enclosed regions) excised 14 days after implantation were analyzed by immunofluorescence for Vegfa (B) and quantified (C) (means ± SEM, $n = 6$ individual plugs per group, $^*p < 0.05$, $^{**}p < 0.01$, $^{***}p < 0.001$, n.s., Student $t$ test). See S1 Data for quantitative data and statistical analysis. See S2 Data for original, uncropped images supporting blots and gel results. n.s., not significant; SerRS, seryl-tRNA synthetase; SerRS^AA, S101A/S241A; SerRS^DD, S101D/S241D; SerRS^WT, wild-type SerRS; Vegfa, vascular endothelial growth factor A

SerRS^AA, or SerRS^DD. Considering that angiogenesis-related transcription factors, such as c-Myc and HIF, are likely to be overexpressed in cancer cells [3,36], we selected MDA-MB-231 cells showing high levels of SerRS overexpression (approximately 7-fold) compared to that of the endogenous protein (S10A Fig). The engineered MDA-MB-231 cells were implanted subcutaneously into the mammary glands of T cell-deficient homozygote NU/J athymic nude (Nude) mice. Strong inhibitory effects of SerRS^WT and SerRS^AA, but not of SerRS^DD, on tumor growth were observed during a 3-week period after the implantation (Fig 6A). Remarkably,

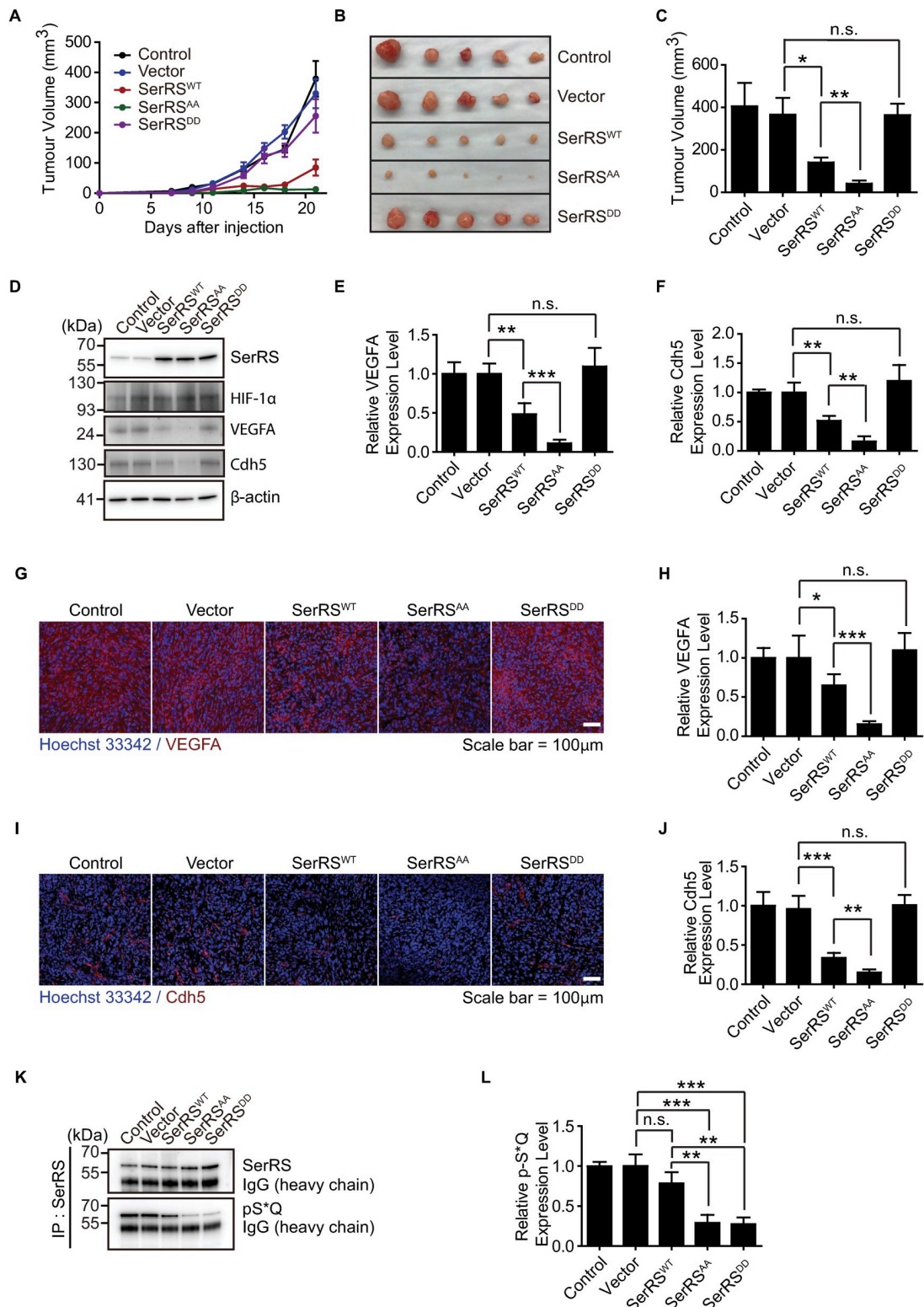

**Fig 6. SerRS^AA exhibits stronger activity than SerRS^WT in inhibiting tumor angiogenesis and tumor growth in mice.** (A–C) Tumor xenografts formed by the original (control) or engineered MDA-MB-231 cells were analyzed by monitoring the growth curves (A) and comparing the size of the xenografts at end point (22 days post implantation) (B) with quantifications (C). (D–F) The dissected MDA-MB-231 breast cancer xenografts were analyzed by western blots for SerRS, HIF-1α, VEGFA, Cdh5, and β-actin protein levels (D), and the levels were quantified for VEGFA (E) and Cdh5 (F). (G–J) Tumor angiogenesis in MDA-MB-

231 xenografts was analyzed by immunofluorescent staining of VEGFA (**G**) and Cdh5 (**I**) and quantified by measuring the fluorescent signal intensities (**H, J**). (**K, L**) IP and western blot analysis to confirm SerRS phosphorylation in tumor xenografts. IP was performed with anti-SerRS antibody from rabbit before phosphorylated SerRS was detected with a specific antibody against p-S*Q (**K**) and quantified by measuring the band intensities (**L**). Data are shown as means ± SEM, $n = 5$ mice, *$p < 0.05$, **$p < 0.01$, ***$p < 0.001$, n.s., not significant, Student $t$ test). See S1 Data for quantitative data and statistical analysis. See S2 Data for original, uncropped images supporting blots and gel results. HIF-1α; hypoxia-inducible factor 1α; IP, immunoprecipitation; n.s., not significant; SerRS, seryl-tRNA synthetase; SerRS[AA], S101A/S241A; SerRS[DD], S101D/S241D; SerRS[WT], wild-type SerRS; VEGFA, vascular endothelial growth factor A.

the phosphor-deficient SerRS[AA] mutant significantly enhanced the inhibitory effect of SerRS[WT]. In fact, SerRS[AA]-expressing MDA-MB-231 cells barely grew into tumors in mice (Fig 6A–6C). At the 3-week end point, the vasculature in tumor xenografts was examined by western blot analysis (Fig 6D–6F) and immunostaining (Fig 6G–6J) to detect VEGFA and endothelial cell marker VE-Cadherin (Cdh5). Consistent with tumor growth, strong inhibitory effects of SerRS[WT] and SerRS[AA], but not of SerRS[DD], were observed. And the effect of SerRS[AA] is significantly stronger than that of SerRS[WT] (Fig 6D–6J). We also detected the level of ATM/ATR-mediated SerRS phosphorylation in tumor tissues. As expected, the level of phosphorylated SerRS was significantly reduced in tumors expressing either SerRS[AA] or SerRS[DD], compared to SerRS[WT] (Fig 6K and 6L), confirming that SerRS phosphorylation at S101 and/or S241 indeed occurs in hypoxic tumor microenvironment (S10B Fig) to inactivate the antiangiogenic function of SerRS to allow sustained tumor growth.

Consistent with the result from the matrigel plug angiogenesis assay (Fig 5), SerRS[WT] exhibited partial, but nevertheless strong, activity in inhibiting tumor angiogenesis and tumor growth (Fig 6). These in vivo observations contrast to the more complete inactivation of SerRS[WT] by hypoxia in cell-based systems (Fig 3G and Fig 4A and 4B). Possibly, the hypoxic environment inside a matrigel plug or a tumor is more heterogenous than in cultured cells, rendering more SerRS[WT] to remain unphosphorylated and able to inhibit VEGFA expression.

## Discussion

Until now, our understanding of angiogenesis regulation during hypoxia has focused on the activation of positive regulators. By studying SerRS, a novel inhibitor of angiogenesis, we revealed the importance of silencing negative regulators in this process. SerRS is inactivated through posttranslational modification by phosphoinositide-3 kinase (PI3K)-like kinase family members ATM and ATR, which are central kinases in DNA damage responses, important for maintaining genome stability under hypoxic stress in mammals [32,37]. ATR is activated with the accumulation of single-stranded DNA generated from stalled DNA replication forks under hypoxia [32]. On the other hand, ATM is activated during reoxygenation, when reactive oxygen species (ROS) are generated and lead to DNA double-stranded breaks [32,37] (Fig 7). Activated ATM/ATR kinases phosphorylate many effector substrates that regulate cell cycle, DNA damage repair, and apoptosis [26,38]. In addition to their role in maintaining genome stability, ATM and ATR, especially ATR, was reported to regulate pathological hypoxia-driven angiogenesis through an unknown mechanism [39]. This report is consistent with our observation that ATM and ATR inhibitors can block hypoxia-induced VEGFA expression (Fig 3H). Moreover, our study identifies SerRS as an ATM/ATR substrate and a novel effector to mediate hypoxia-driven angiogenesis. The facts that SerRS is a negative regulator of angiogenesis and that the activity of SerRS is inactivated by ATM/ATR reveal a general principle that hypoxia-driven angiogenesis requires not only the activation of positive regulators, such as HIF and c-Myc, but also the silencing of negative regulators, such as SerRS (Fig 7).

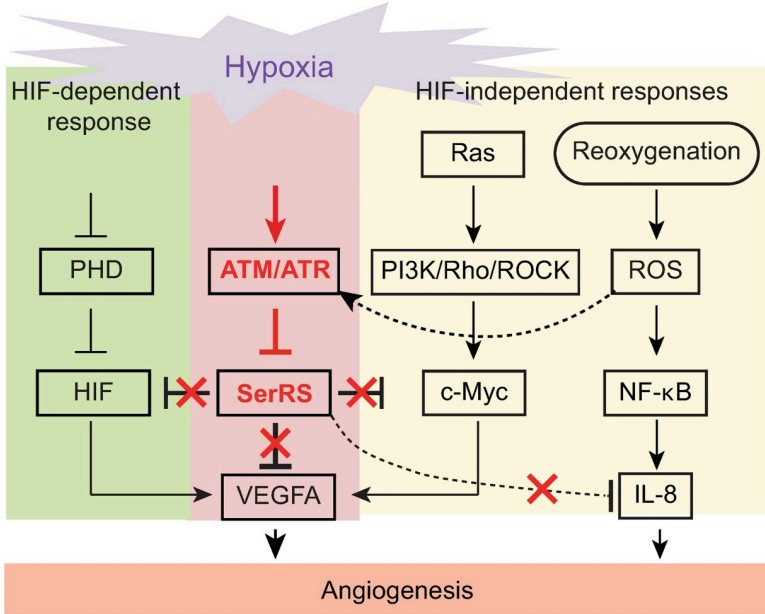

**Fig 7. Schematic diagram to illustrate the impact of SerRS in hypoxia-induced angiogenesis.** Multiple pathways are activated under hypoxia to induce angiogenesis. SerRS has the capacity to inhibit these pathways regardless of whether they are dependent or independent of HIF. This capacity, however, is silenced by ATM/ATR, which is also activated by hypoxia, highlighting the central role of SerRS in regulating angiogenesis. ATM, ataxia telangiectasia mutated; ATR, ataxia telangiectasia mutated and RAD3-related; HIF; hypoxia-inducible factor; SerRS, seryl-tRNA synthetase.

Tumor cells can adapt to the hypoxic environment through orchestrating the expression of a large number of genes that functions in apoptosis, cell survival, energy metabolism, and angiogenesis [3]. Although HIF is considered as the most important transcription factor mediating the hypoxia response and has been targeted in many preclinical and clinical anticancer programs, success has been limited. In fact, HIF is not always associated with increased cancer mortality, and overexpression of HIF does not always occur in solid tumors [40]. In some cases, inhibition of HIF only delays tumors growth, but tumors growth eventually resumes in an HIF-independent manner [1].

For example, Kirsten rat sarcoma viral oncogene homolog (K-RAS) and hypoxia tension can synergistically increase the level of guanosine 5′-triphosphate (GTP)-bound Rho via PI3K and lead to the phosphorylation of c-Myc, which promotes VEGFA expression in the absence of HIF-1 [16] (Fig 7). Another HIF-independent angiogenesis pathway has been observed in HIF-1-deficient colon cancer cells, where increased production of ROS under hypoxia activates nuclear factor kappa B (NF-κB), which strongly induces angiogenic factors such as interleukin 8 (IL-8) [15,17] (Fig 7).

Our previous studies have revealed that SerRS strongly competes with c-Myc for binding to the *VEGFA* promoter and represses VEGFA expression [19]. Consistently, here we show that phosphorylation-mediated loss of DNA binding of SerRS under hypoxia is required for c-Myc to bind to the *VEGFA* promoter (Fig 3D and 3E). Remarkably, SerRS can also compete with HIF-1 for binding to the DNA (Fig 3D and 3E), indicating that SerRS inhibits both HIF-dependent and HIF-independent activation of VEGFA. Interestingly, competitive binding between SerRS and an NF-κB family protein on *VEGFA* was also reported [41]. Moreover, we have observed a stimulation of IL-8 expression (among 215 angiogenesis related genes) in human endothelial cells when SerRS is knocked down (S11 Fig). Although it is unclear whether IL-8 is a direct target of SerRS, the apparent effect suggests that, in addition to

inhibiting VEGFA expression through blocking c-Myc and HIF-1, SerRS may also inhibit IL-8 expression and potentially antagonize the ROS/NF-κB/IL-8 hypoxia response pathway (Fig 7). Therefore, we propose that inactivation of SerRS through ATM/ATR is required for the activation of several HIF-dependent and HIF-independent pathways for hypoxia-driven angiogenesis. Hence, blocking the effect of ATM/ATR on SerRS, for example, through the expression of a phosphorylation-deficient form of SerRS, can act as a master inhibitor of multiple hypoxia response pathways to effectively inhibit angiogenesis. Our results with the endothelial cell tube formation assay (Fig 4C and 4D) and in mice showing potent antiangiogenesis and antitumor effect with SerRS[AA] expression (Figs 5 and 6) are supportive of this concept.

In conclusion, our study identified a key regulation step in driving angiogenesis under hypoxia, which involves the inactivation of a seemingly broad functioning negative regulator of angiogenesis. Notably, this regulator is a nuclear localized tRNA synthetase, highlighting the importance of the non-translation function of a translation factor in angiogenesis regulation.

## Material and methods

### Cell lines

Human HEK293 and MDA-MB-231 and mouse 3B11 cell lines were purchased from American Type Culture Collection (ATCC, Manassas, Virginia, United States of America) and cultured in Dulbecco's Modified Eagle Medium (ThermoFisher Scientific, Grand Island, New York, USA) supplemented with heat-inactivated fetal bovine serum (Omega Scientific, Tarzana, California, USA) to a final concentration of 10%. HUVEC cells were purchased from ATCC and cultured in EBM basal medium supplemented with EGM Endothelial Cell Growth Medium SingleQuots Supplements (Lonza, Allendale, New Jersey, USA) and heat-inactivated fetal bovine serum (Biological Industries, Israel) to a final concentration of 8%. Transient transfections were performed using Lipofectamine 2000 (ThermoFisher Scientific). We established 3B11 and MDA-MB-231 cell lines stably expressing mouse and human SerRS mutants, respectively, by using pBabe-puro (Addgene, Cambridge, Massachusetts, USA) vector-based retroviral infections and selection with puromycin (Sigma-Aldrich, St. Louis, Missouri, USA). The coding sequences for wild-type (WT) SerRS and its mutants (AA and DD) were subcloned into lentiviral vector pLV-EF1α-MCS-IRES-Bsd (Biosettia, San Diego, California, USA), and the lentivirus were prepared to infect HUVEC cells. The stably transfected HUVEC cells were selected by 2.5 μg/mL of blasticidin. Hypoxia was induced using an enzymatic hypoxia system [35]. Hypoxia medium was prepared by rapidly diluting glucose oxidase and catalase at a constant 1:10 ratio in cell culture medium (Sigma-Aldrich, #C3155 and #G0543), the concentration of the 2 enzymes are as follows: glucose oxidase (2 U/ml) and catalase (120 U/ml). For incubation periods longer than 24 h, the medium was replaced by preequilibrated hypoxic medium to maintain nutrients and substrates such as glucose.

### Plasmid constructs

Human and mouse full-length SerRS genes were cloned into the pFlag-CMV2 vector (Sigma-Aldrich) and pBabe-puro vector (Addgene) and human SIRT2 genes into the pcDNA6c-V5/His vector (Thermo Fisher Scientific). For mutations in SerRS, we performed site-directed mutagenesis PCR to obtain the SerRS[AA] and SerRS[DD] construct. The primer sequences for human SerRS mutant constructs are 5′-GAA AGT CGC ACA AAT CAA AAA AGT CCG ACT CCT CAT TG-3′ and 5′-TGA TTT GTG CGA CTT TCA GGT TAG CTA AAG CGT C-3′ for S101A; 5′-GAA AGT CGA CCA AAT CAA AAA AGT CCG ACT CCT CAT TG-3′ and 5′-TGA TTT GGT CGA CTT TCA GGT TAG CTA AAG CGT C-3′ for S101D; 5′-AGC TCG CAC AGT TTG ATG AAG AAC TTT ATA AGG-3′ and 5′-AAC TGT GCG AGC TGT GCC

ACC TCC TGC ATG ACC TCC-3′ for S241A; 5′-AGC TC<u>G AC</u>C AGT TTG ATG AAG AAC TTT ATA AGG-3′ and 5′-AAC TG<u>G TC</u>G AGC TGT GCC ACC TCC TGC ATG ACC TCC-3′ for S241D. The primer sequences for mouse SerRS mutant constructs are 5′-GAA AGT C<u>GC A</u>CA GAT TAA AAA AGT CCG ACT CCT CAT TG-3′ and 5′-TAA TCT G<u>TG C</u>GA CTT TCA GGG CAG CTA GCG CGT C-3′ for S101A; 5′-GAA AGT C<u>GA C</u>CA GAT TAA AAA AGT CCG ACT CCT CAT TG-3′ and 5′-TAA TCT G<u>GT C</u>GA CTT TCA GGG CAG CTA GCG CGT C-3′ for S101D; 5′- CAG CTC <u>GCC</u> CAG TTT GAT GAA GAA CTT TAT AAG GTG-3′ and 5′-CAA ACT G<u>GG C</u>GA GCT GGG CCA CTT CCT GCA TG-3′ for S241A; 5′-CAG CTC <u>GAC</u> CAG TTT GAT GAA GAA CTT TAT AAG GTG-3′ and 5′-CAA ACT G<u>GT C</u>GA GCT GGG CCA CTT CCT GCA TG-3′ for S241D. The underlined sequences encode the substituted residues.

For protein purification, human SerRS and its mutant genes were subcloned into pET-20b (+) plasmid (Novagen, Darmstadt, Germany) and overexpressed in *E. coli*. The recombinant carboxyl-terminal His$_6$-tagged proteins were purified using Ni-NTA beads (Qiagen, Valencia, California, USA). The purities of the recombinant proteins were assessed by Coomassie blue staining following 4% to 12% Mini Gel (ThermoFisher Scientific) electrophoresis. Protein concentrations were determined using Bradford protein assay (BioRad, Hercules, California, USA).

## RNAi

DNA oligos encoding shRNAs designed against human SerRS (5′-GGC ATA GGG ACC CAT CAT TGA-3′ in 3′UTR), GlyRS (5′-GCA TGG AGT ATC TCA CAA AGT-3′ in the open reading frame), were inserted into the pLentiLox-hH1 plasmid, modified from the pLentiLox 3.7 plasmid to contain an H1 promoter (between Xba I and Xho I sites) to drive the shRNA expression. For nontargeting control shRNA, we used the sequence 5′-TAA GGC TAT GAA GAG ATA C-3′. Small interfering RNA (siRNA) duplexes against ATM and ATR were purchased from Cell Signaling Technology (Danvers, Massachusetts, USA). Cells were transfected with shRNA plasmids or siRNA duplexes by using Lipofectamine 2000 reagent (ThermoFisher Scientific). Forty-eight hours post-transfection, cells were subjected to hypoxia treatment and analysis such as real-time quantitative reverse transcription PCR (qRT-PCR) and western blot. Lentivirus-based RNA interference (RNAi) vectors against ATM and ATR were made using pLV-H1-EF1α-puro RNAi plasmid (#SORT-B19, BiOSETTIA, San Diego, California, USA). The sequences for shRNA templates are as follows: shATM#1 (5′-AAA AGC TGC AGA GTC AAT CAA TAG ATT GGA TCC AAT CTA TTG ATT GAC TCT GCA GC-3′), shATM#2 (5′-AAA AGC AGA AAC ACT CCC AGC TTC TTT GGA TCC AAA GAA GCT GGG AGT GTT CTG C-3′), shATR #1 (5′- AAA AGC CGC TAA TCT CTT AAC ATT ATT GGA TCC AAT AAT GTT AGA AGA TTA GCG GC 3′), shATR#2 (5′-AAA AGC ATA TAC ACA TGC CCA AAT ATT GGA TCC AAT ATT TGG GCA TGT GTA TAT GC-3′).

## Real-time PCR assay

Total RNA was isolated from cells by TRIzol Reagent (ThermoFisher Scientific). One microgram (μg) of the total RNA from each sample was reversely transcribed to cDNA by M-MLV reverse transcriptase (Promega, Madison, Wisconsin, USA). All real-time PCR reactions were performed using the StepOnePlus Real-Time PCR system (ThermoFisher Scientific) with SYBR Select Master Mix (Applied Biosystems). The primer pairs for the PCR reactions were: 5′-GAG GGC AGA ATC ATC ACG AAG-3′ and 5′-TGT GCT GTA GGA AGC TCA TCT CTC-3′ for human *VEGFA*; 5′-CGT CAC CAA CTG GGA CGA-3′ and 5′-ATG GGG GAG GGC ATA CC-3′ for human *β-ACTIN*; 5′- GGC TCT CCT CCA TCT GTC TGC-3′ and 5′-

CAG TGG TTT TCT TTC TTT GCT TTG-3′ for zebrafish *vegfa*; 5′-TCA CCA CCA CAG CCG AAA GAG-3′ and 5′-GTC AGC AAT GCC AGG GTA CAT-3′ for zebrafish *β-actin*. The PCR reaction program started at 95˚C for 10 min, followed by 45 cycles of 95˚C for 20 s and 60˚C for 1 min. Each experiment was carried out in triplicate. The *VEGFA* gene expression was normalized to that of *β-ACTIN*. Statistical analyses were performed with the software SigmaPlot (version 10.0). Student *t* test was used to analyze the changes between different groups.

## $^{32}$P-labeling assay using nuclear extract

The in vitro phosphorylation of SerRS by ATM/ATR was performed based on the methods described by Schiotani and colleagues [29]. Briefly, to prepare the nuclear extract, $5 \times 10^7$ HEK293 cells were treated with swelling buffer [10 mM HEPES–NaOH (pH 7.9), 10 mM KCl, 1.5 mM MgCl$_2$, 0.5 mM DTT, 0.5 mM PMSF, and protease inhibitor cocktail (added before use)] and homogenized to remove the cytoplasmic fraction. The nuclear pellets were lysed in lysis buffer [20 mM HEPES–NaOH (pH 7.9), 600 mM KCl, 1.5 mM MgCl$_2$, 0.2 mM EDTA, 25% glycerol, 0.5 mM DTT, 0.5 mM PMSF, and protease inhibitor cocktail (added before use)], and the supernatant was dialyzed in the storage buffer [20 mM HEPES–NaOH (pH 7.9), 100 mM KCl, 0.2 mM EDTA, 20% glycerol, 0.5 mM DTT, and 0.5 mM PMSF].

For the in vitro $^{32}$P-labeling with activated ATM/ATR, in a 20 μL reaction system, 10 to 20 μg of recombinant human SerRS or GlyRS proteins with His$_6$-tag were mixed with 5 μL of 4× Reaction buffer [40 mM HEPES–KOH (pH 7.6), 200 mM KCl, 4 mM PMSF, 2 mM DTT, and protease inhibitor cocktail], 4 μL of 5× ATP buffer [5 mM ATP, 0.5 mM MgCl$_2$, 50 μg/mL Creatine kinase, and 25 mM phosphocreatine], 5 μL of Nuclear extract, 2 μL of double-stranded DNA oligos [the sequence of the plus strand: 5′-TGCAGCTGGCACGACAGGTTT-TAATGAATCGGCCAACGCGCGGGGAGAGGCGGTTTGCGTATTGGGCGCT-3′, 50 ng/μL final concentration], 1 μL of γ-32P-ATP [3,000 Ci/mmol at 10 mCi/mL] and incubate in 37˚C for 30 min. Recombinant proteins were then purified by Ni-NTA beads and subjected to SDS-PAGE and autoradiography.

## Immunoblotting and immunoprecipitation

Cells were resuspended with lysis buffer [20 mM Tris-HCl (pH 7.5), 150 mM NaCl, 1 mM of EDTA, 1 mM EGTA, 1% Triton X-100, 2.5 mM sodium pyrophosphate, 1 mM β-glycerophosphate, 1 mM Na$_3$VO$_4$, and protease inhibitor cocktail] on ice. Supernatants were incubated with indicated antibodies and protein-G-conjugated agarose beads (ThermoFisher Scientific) for at least 2 h. The beads were washed 5 times with wash buffer (same as the lysis buffer, except that Triton X-100 was reduced from 1% to 0.1%) and then subjected to SDS-PAGE and immunoblotting analysis with indicated antibodies. The monoclonal anti-Flag antibody was purchased from Sigma-Aldrich (#F1804). Custom-made rabbit antihuman SerRS antibody was raised against purified human recombinant SerRS and affinity-purified. In HEK293 cells, phosphorylated SerRS was detected with the rabbit polyclonal antibody against phosphor-ATM/ATR substrate (p-S*Q, #6966, Cell Signaling Technology) after IP by mouse monoclonal anti-SerRS antibody (1H4, #WH0006301M1, Sigma-Aldrich). For the detection of hypoxia-induced SerRS phosphorylation in HUVEC cells and in human tumor xenograft tissues, a rabbit polyclonal anti-SerRS antibody (Custom-made at Scripps) was used to IP SerRS, followed by western blot using a rabbit antibody against phosphor-ATM/ATR substrate (p-S*Q) and a rabbit antibody against SerRS (Custom-made by Genway, San Diego). The anti-ATM (#2873), anti-p-ATM (serine 1981, #13050), anti-ATR (#2790), anti-SIRT2 (#21672), anti-α-tubulin (#3873), anti-β-actin (#3700), anti-LMNA (#2032), anti-P53 (#9282), anti-p-P53 (serine 15, #9284), anti-RPA32 (#52448), anti-CHK1 (#2360), anti-p-CHK1 (serine 345, #2348), anti-

CHK2 (#2662), and anti-p-CHK2 (threonine 68, #2661) antibodies were purchased from Cell Signaling Technology. Anti-p-RPA32 (serine 33, #ab211877) antibody was purchased from Abcam (Cambridge, Massachusetts, USA). Anti-HIF-1α antibody (#NB100-105) was purchased from Novus Biologicals (Littleton, Colorado, USA). Anti-V5 (#R960CUS) and anti-GlyRS (#PAB28573) antibodies were purchased from ThermoFisher Scientific and Abnova (Walnut, California, USA), respectively. Anti-VEGF antibody (VEGF(c1), #SC-7269) was purchased from Santa Cruz (Santa Cruz, California, USA).

## In vivo studies in zebrafish

Transgenic *Tg* (*fli1a*: *EGFP*) fish were maintained as described before [20]. The fish embryos were kept at 28.5˚C before and after microinjection. The antisense morpholino (MO) targeting SerRS was injected into the yolk of 1-cell stage embryos at the dosage of 4~5 ng per embryo. The sequences of the MOs are as follows: SerRS-MO: 5′-AGG AGA ATG TGA ACA AAC CTG ACA C-3′ [20] and the standard control MO: 5′-CCT CTT ACC TCA GTT ACA ATT TAT A-3′ (purchased from Gene Tools LLC, Philomath, Oregon, USA). After injection, embryos were incubated in E3 embryo medium supplemented with 0.003% 1-phenyl-2-thiourea (PTU) at 28.5˚C to prevent pigment formation. Embryos were anesthetized with 0.168 mg/mL tricaine (Sigma-Aldrich), mounted in 2% methylcellulose and photographed with a Nikon fluorescent microscope (AZ100) equipped with a Nikon CCD camera (QimagingRetiga 2000R). All the experiments involving zebrafish had been conducted according to the guidelines established by the Institutional Animal Care and Use Committee (IACUC) at The Scripps Research Institute, IACUC approval number 09–0009. Statistical analyses were performed with the software SPSS Statistics 19. The rescue effects of different SerRS mutants on ISV development were analyzed with $\chi^2$ test.

## Matrigel plug angiogenesis assay

A total of $10^6$ of stably transfected 3B11 cells were resuspended in 100 μL of DMEM medium and then mixed with 200 μL of ice-cold matrigel (BD Biosciences, San Jose, California, USA) liquid on ice. The 300 μL of cell and matrigel mixture was injected subcutaneously into the flank of C3H/HeJ mice (2 injection sites per mouse and 5 to 6 mice in each group) (Jackson Laboratory). Fourteen days after inoculation, the matrigel plugs were excised and frozen in Tissue-Tek OCT compound for cryostat sectioning. All the mouse experiments were conducted according to the guidelines established by the IACUC (protocol number 13–0003) at The Scripps Research Institute.

## Xenograft tumor model

A total of $10^6$ of MDA-MB-231 cells stably overexpressing WT human SerRS, SerRS$^{AA}$, or SerRS$^{DD}$ were subcutaneously injected into the mammal glands of 6- to 8-week-old female NU/J mice (5 mice in each group) (Jackson Laboratory, stock #002019). Twenty-two days after injection, the tumor xenografts were separated from mice and frozen in Tissue-Tek OCT compound for cryostat sectioning.

## Immunohistochemistry and immunofluorescence

Sections of 5 μm in thickness from freshly frozen tumors xenograft and matrigel plugs were treated with acetone and 3% $H_2O_2$ to block endogenous peroxidase. After 3 to 5 times wash and goat serum block, the sections were incubated with an anti-CD31 antibody (diluted at 1:3000; Cell Signaling Technology, #3528) overnight at 4˚C. The blood vessels were counted in

5 to 10 random viable fields (magnification, 400×) in tumor xenograft samples, and microvessel density in matrigel plugs was quantified by measuring the CD31 immunostaining density using Image J software. To detect the hypoxic regions, we incubated the slides with anti-HIF-1α antibody (1:100; Novus Biologicals, #NB100-105). The immunofluorescence staining was performed with anti-VEGFA antibody (diluted at 1:500; Abcam, #ab46154) and anti VE-cadherin (Cdh5) antibody (diluted at 1:500, Abcam, #ab33168).

## EMSA

The 27-bp DNA oligonucleotides corresponding to SerRS binding site on the *VEGFA* promoter (5′-GGC GGG GCG GAG CCA TGC GCC CCC CCC-3′) were synthesized, annealed, and [$^{32}$P]-labeled at the 5′ end by T4 polynucleotide kinase (New England Biolabs, Ipswich, Massachusetts, USA) before desalting using a sephadex G-25 spin column (GE Healthcare, Pittsburgh, Pennsylvania, USA). The labeled oligonucleotides (0.08 pmol) were incubated with recombinant SerRS at indicated concentrations in binding buffer [20 mM Tris-HCl (pH 8.0), 60 mM KCl, 5 mM MgCl$_2$, 0.1 mg/mL BSA, 10 ng/μL poly (dG-dC), 1 mM DTT] for 30 min at room temperature. The samples were loaded to 5% native polyacrylamide gel (17.5 cm in length) and underwent electrophoresis at 250 V in running buffer (25 mM Tris (pH 8.3), 190 mM glycine). Afterwards, the gel was dried and examined by autoradiography.

## Cell fractionation

Cytoplasmic and nuclear fractions were separated and extracted using NE-PER Nuclear and Cytoplasmic Extraction Kit (ThermoFisher Scientific). Exogenously or endogenously expressed SerRS proteins were detected by western blot analysis using anti-flag polyclonal antibody (Sigma-Aldrich) or polyclonal anti-SerRS antibody.

## Chromatin immunoprecipitation (ChIP)

Stably transfected HEK293 or HUVEC cells were fixed with formaldehyde (1% final concentration) for 10 min at room temperature. The reaction was stopped by adding 125 mM of glycine. ChIP assays were performed according to the protocol of ChIP-IT Express Enzymatic kit (Active Motif) with affinity purified polyclonal anti-SerRS antibody (custom-made), anti-c-Myc antibody (Cell Signaling Technology, #9402), and anti-HIF-1α antibody (Cell Signaling Technology, #36169). After 3 washes, ChIPed DNA was analyzed on the StepOnePlus Real-Time PCR system using SYBR Select Master Mix (Applied Biosystems). The primer set (forward: 5′-GGGCGGATGGGTAATTTTCA-3′, reverse: 5′- CTGCGGACGCCCAGTGAA- 3′) targeting the *VEGFA* promoter was used.

## ELISA

After the stably transfected HUVEC or HEK293 cells were cultured under normoxia or hypoxia for 12 h, the level of secreted VEGFA proteins in cell culture medium were detected by Human VEGFA ELISA Kit (Wuxi Donglin Sci&Tech Development, Jiangsu, China).

## Aminoacylation assay

As described previously [42], the aminoacylation assays were performed at room temperature with 50 nM enzyme in 50 μL reaction mixture contains 50 mM Hepes (pH 7.5), 20 mM KCl, 5 mM MgCl$_2$, 4 mM ATP, 2 mM DTT, 5 μg/mL pyrophosphatase, 20 μM cold L-serine, 1.34 μM [3H]-serine (1 mCi/mL), and 200 μM yeast total tRNA (Roche Diagnostics, Indianapolis, Indiana, USA). The reaction was initiated by adding 50 nM enzyme into the reaction mixture.

Aliquots of 5 μL were applied into 100 μL quench solution (300 mM NaOAc (pH 3.0), 1 mg/mL DNA, and 100 mM EDTA) in the Multi Screen 96-well filter plate (0.45 μm pore size hydrophobic, low-protein-binding membrane; Merck Millipore) at different time points of the reaction. After that, 100 μL of 20% (w/v) trichloroacetic acid (TCA) was added to precipitate the nucleic acids in each well. Plate was then washed 4 times with 200 μL/well of 5% TCA containing 100 mM cold L-serine, followed by 1-time wash with 95% ethanol. Air-dried plate was then treated with 70 μL/well of 100 mM NaOH to elute tRNAs. The eluted tRNAs were centrifuged into a 96-well flexible PET microplate (PerkinElmer, Santa Clara, California, USA) with 150 μL/well of Supermix scintillation mixture. After mixing, the radioactivity in each well of the plate was measured in a PerkinElmer 1450 Liquid Scintillation Counter and Luminescence Counter.

## Tube formation assay

HUVEC cells ($5 \times 10^5$) were suspended in fresh EGM medium mixed with same volume of conditional medium from empty vector, or SerRS$^{WT}$, or SerRS$^{AA}$, or SerRS$^{DD}$ stably transfected HUVEC cells, which were cultured in normoxia or hypoxia conditions for 12 h, and seeded on matrigel basement membrane matrix (BD Biosciences) in 48-well plates. Thirty-six hours later, the images of the endothelia cell tubular network were taken with a Q-IMAGING MicroPublisher 5.0 RTV CCD camera attached to an inverted Olympus IX71 microscope. The length of the tubes was measured by Image J software. To test the contribution of VEGFA, a VEGFA neutralizing antibody (Sino Biological, Beijing, China, #11066-R010-500) was added to the medium at the final concentration of 100 ng/mL.

## Statistics

Statistical analysis was done either by χ2 test or Student $t$ test using GraphPad Prism (GraphPad Software, San Diego, California, USA). n.s., not significant; $^*$ $p < 0.05$; $^{**}$ $p < 0.01$; $^{***}$ $p < 0.001$; $^{****}$ $p < 0.0001$.

## Supporting information

**S1 Fig. SerRS is phosphorylated by ATM/ATR kinases at S101 and S241 under hypoxia.** (A) Western blot to monitor gene knockdown efficiencies by RNAi in HEK293 cells transfected with sh-SerRS, sh-GlyRS, or nonspecific sh-Control under normoxic and hypoxic (12 h) conditions. (B) Western blot to show hypoxia does not affect the protein level of SerRS (endogenous) in HEK293 cell. As expected, the HIF-1α level is increased under hypoxia. (C) In vitro $^{32}$P-labeling assay to confirm S101 and S241 as the phosphorylation sites on SerRS. (D) Hypoxia-induced SerRS phosphorylation is inhibited by specific ATM inhibitor KU-55933 and ATR inhibitor VE-821. Phosphorylated SerRS was detected with the antibody against phosphor-ATM/ATR substrate (p-S$^*$Q) followed by immunoprecipitation with the anti-SerRS antibody. Activation of ATM and ATR was confirmed by phosphorylation of CHK1 and CHK2. See S2 Data for original, uncropped images supporting blots and gel results. ATM, ataxia telangiectasia mutated; ATR, ataxia telangiectasia mutated and RAD3-related; HIF-1α; hypoxia-inducible factor 1α; RNAi, RNA interference; SerRS, seryl-tRNA synthetase; sh-Control, control shRNA; sh-GlyRS, shRNAs targeting GlyRS; sh-SerRS, shRNAs targeting SerRS. (EPS)

**S2 Fig. Hypoxia generated by the 2-enzyme method does not induce DNA damage.** HEK293 cells were cultured in normoxia or hypoxia generated by adding glucose oxidase (2 U/ml) and catalase (120 U/ml) in the medium for indicated durations before analysis for

potential DNA damage by western blot against phosphorylated H2AX ($\gamma$-H2AX). Cells treated with UV (12 mJ/cm$^2$) were used as the positive control. See S2 Data for original, uncropped images.
(EPS)

**S3 Fig. SerRS phosphorylation-mimicry mutant (SerRS$^{DD}$) can no longer support normal vascular development in zebrafish.** The activities of SerRS$^{WT}$, SerRS$^{AA}$, and SerRS$^{DD}$ in regulating vascular development were examined in zebrafish injected with SerRS-MO and with co-injection of SerRS-MO and SerRS$^{WT}$, SerRS$^{AA}$, or SerRS$^{DD}$ mRNA. The abnormal hyper-vascularization phenotype was indicated by red arrows (A) and quantified (B, $n = 125$–211 per group, ****$p < 0.0001$, n.s., $\chi$2 test). n.s., not significant; SerRS, seryl-tRNA synthetase; SerRS$^{AA}$, S101A/S241A; SerRS$^{DD}$, S101D/S241D; SerRS-MO, antisense morpholino against SerRS; SerRS$^{WT}$, wild-type SerRS.
(EPS)

**S4 Fig. SerRS phosphorylation at S101 and S241 does not affect nuclear localization and SerRS-SIRT2 interaction.** (A) Cell fractionation to evaluate the effect of hypoxia on nuclear localization of SerRS. The Cy fractions, Nu fractions, and the WCLs of HEK293 cells with and without hypoxia treatment were examined by western blot with antibodies against SerRS, nuclear protein LMNA, and cytosolic protein $\alpha$-tubulin. (B) Cell fractionation to evaluate the effect of phosphorylation on SerRS nuclear localization. HEK293 cells were transfected with Flag-tagged SerRS$^{WT}$, SerRS$^{AA}$, or SerRS$^{DD}$ constructs and subjected to cell fractionation. SerRS nuclear localization was examined by western blot with antibodies against the Flag tag, nuclear protein LMNA, and cytosolic protein $\alpha$-tubulin. (C) Co-immunoprecipitation to examine the interaction between endogenous SerRS and SIRT2 in HEK293 cells with and without hypoxia stress. (D) Co-immunoprecipitation to examine the interaction between exogenously expressed SerRS (WT and mutants, Flag-tagged) and SIRT2 (V5-tagged) in HEK293 cells. See S2 Data for original, uncropped images. Cy, cytosolic fractions; LMNA, Lamin A/C; Nu, nuclear fractions; SerRS, seryl-tRNA synthetase; SerRS$^{AA}$, S101A/S241A; SerRS$^{DD}$, S101D/S241D; SerRS$^{WT}$, wild-type SerRS; SIRT2, sirtuin 2; WCL, whole cell lysates.
(EPS)

**S5 Fig. In vitro tRNA aminoacylation assay to evaluate the mutational impact on SerRS enzymatic activity.** The AA and DD mutations do not significantly affect the enzymatic activity of SerRS. SerRS$^{WT}$ (black) and SIRT2 (green) are used as positive and negative controls, respectively, for the assay. See S1 Data for numerical data and quantitative analysis. SerRS, seryl-tRNA synthetase; SerRS$^{WT}$, wild-type SerRS; SIRT2, sirtuin 2.
(EPS)

**S6 Fig. SerRS does not interact with c-Myc and HIF-1$\alpha$ proteins.** Co-immunoprecipitation analysis could not detect any interaction between SerRS (WT, AA, or DD mutants, Flag-tagged) and c-Myc or HIF-1$\alpha$ under normoxic or hypoxic conditions. See S2 Data for original, uncropped images. HIF-1$\alpha$; hypoxia-inducible factor 1$\alpha$; SerRS, seryl-tRNA synthetase.
(EPS)

**S7 Fig. SerRS-regulated VEGFA induction by hypoxia depends on its binding to the VEGFA promoter.** (A) Schematics of WT promoter ($-1,005$~$+379$) of human VEGFA gene with HIF-1$\alpha$/HIF-1$\beta$ binding site, SBS, and c-Myc binding site indicated and the SBS-deleted promoter ($\Delta$SBS). (B) Luciferase reporter analysis on HEK293 cells transfected with firefly luciferase reporters driven by WT or $\Delta$SBS VEGFA promoters under normoxic or hypoxic conditions with or without SerRS$^{AA}$ expression. Data are shown as means ± SEM, $n = 4$,

biological replicates, $^{**}p < 0.01$, $^{***}p < 0.001$, unpaired Student $t$ test. See S1 Data for quantitative data and statistical analysis. HIF-1α; hypoxia-inducible factor 1α; HIF-1β; hypoxia-inducible factor 1β; SBS, SerRS binding site; SerRS, seryl-tRNA synthetase; SerRS$^{AA}$, S101A/S241A; VEGFA, vascular endothelial growth factor A; WT,wild-type.
(EPS)

**S8 Fig. ATM and ATR are essential for hypoxia-induced VEGFA expression.** (A) Western blot analysis of ATM and ATR in HEK293 cells transfected with shATM or shATR or both shRNAs. shlacZ serves as a negative control. (B) qRT-PCR analysis of VEGFA induction under hypoxia in HEK293 cells transfected with indicated shRNAs. Data are shown as means ± SEM, $n = 3$, biological replicates, $^{*}p < 0.05$, $^{**}p < 0.01$, $^{***}p < 0.001$, unpaired Student $t$ test. See S1 Data for quantitative data and statistical analysis. See S2 Data for original, uncropped images. ATM, ataxia telangiectasia mutated; ATR, ataxia telangiectasia mutated and RAD3-related; qRT-PCR, real-time quantitative reverse transcription PCR; shATM, shRNAs against ATM; shATR, shRNAs against ATR; shlacZ, shRNA targeting bacterial lacZ gene; shRNA, short hairpin RNA; VEGFA, vascular endothelial growth factor A.
(EPS)

**S9 Fig. Matrigel plug angiogenesis assay with mouse 3B11 cells.** (A) Immunohistochemistry analysis using anti-HIF-1α antibody to confirm the hypoxic environment in the matrigel plugs (encircled in dash lines). (B, C) Matrigel plug angiogenesis assay performed with stably transfected 3B11 cells in C3H/HeJ mice. Matrigel plugs (dash lines enclosed regions) excised 14 days after implantation were analyzed by immunohistochemistry staining for CD31 (B) and the quantification (C). Data are shown as means ± SEM, $n = 10$–11 individual plugs for each group, $^{***}p < 0.001$, Student $t$ test. See S1 Data for quantitative data and statistical analysis.
(EPS)

**S10 Fig. Overexpression of SerRS in tumor xenografts of the engineered MDA-MB-231 cells.** (A, B) WB to measure the expression of SerRS proteins (A) and HIF-1α (B) in MDA-MB-231 xenografts stably transfected with tag-free human SerRS$^{WT}$ and mutants (SerRS$^{AA}$ and SerRS$^{DD}$). The protein levels were quantified by the density of the WB bands and indicated relative to the level of the endogenous SerRS (Control) (means ± SEM, $n = 5$ mice in each group, $^{*}p < 0.05$, $^{**}p < 0.01$, $^{***}p < 0.001$, Student $t$ test). See S1 Data for quantitative data and statistical analysis. HIF-1α; hypoxia-inducible factor 1α; SerRS, seryl-tRNA synthetase; SerRS$^{AA}$, S101A/S241A; SerRS$^{DD}$, S101D/S241D; SerRS$^{WT}$, wild-type SerRS; WB, western blot.
(EPS)

**S11 Fig. SerRS suppresses the expression of IL-8 among many pro-angiogenic genes.** (A) qRT-PCR analysis of SerRS in HUVEC cells stably transfected with shSerRS or nonspecific shControl. Data are shown as means ± SEM, $n = 3$, biological replicates, $^{***}p < 0.001$, unpaired Student $t$ test. (B, C) qPCR-based microarray assay to examine the changes in the expression of 215 angiogenesis-related genes (B) in SerRS-silenced HUVEC cells (shSerRS) versus control HUVEC cells (shControl). Genes with over 2-fold change in expression resulting from SerRS knockdown were plotted (C). Data are shown as mean values, $n = 4$, biological replicates. The gene expression analysis was performed using the Smart-Chip Real-Time PCR system of WaferGen Biosystems. Angiogenesis-related genes that are repressed or induced upon SerRS knockdown are highlighted in blue and red boxes, respectively. See S1 Data for quantitative data and statistical analysis. HUVEC, human umbilical vein endothelial cell; IL-8, interleukin 8; qPCR, quantitative real-time PCR; qRT-PCR, real-time quantitative reverse transcription PCR; SerRS, seryl-tRNA synthetase; shControl, control shRNA; shSerRS, shRNAs targeting

SerRS.
(EPS)

**S1 Data. Original numerical data and quantitative analysis supporting all figures in this article.**
(XLSX)

**S2 Data. Original, uncropped images supporting all blot and gel results in this article.**
(PPTX)

## Acknowledgments

We thank Professor Paul Schimmel for valuable discussions and Drs. Shanshan Lian and Delgado Valdez for experiments confirming the effect of SerRS phosphorylation in zebrafish.

## Author Contributions

**Conceptualization:** Yi Shi, Xiang-Lei Yang.

**Data curation:** Yi Shi, Ze Liu, Qian Zhang, Ingrid Vallee, Zhongying Mo, Shuji Kishi.

**Formal analysis:** Yi Shi, Ze Liu, Qian Zhang.

**Funding acquisition:** Yi Shi, Xiang-Lei Yang.

**Investigation:** Yi Shi, Ze Liu, Qian Zhang, Ingrid Vallee, Zhongying Mo.

**Methodology:** Yi Shi, Ze Liu, Qian Zhang, Shuji Kishi.

**Project administration:** Yi Shi, Xiang-Lei Yang.

**Software:** Yi Shi, Ze Liu.

**Supervision:** Xiang-Lei Yang.

**Validation:** Yi Shi, Ze Liu.

**Writing – original draft:** Yi Shi, Xiang-Lei Yang.

**Writing – review & editing:** Yi Shi, Ze Liu, Xiang-Lei Yang.

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
