## [Editor Report · Decision Letter 0]

1 Dec 2019

Dear Dr Yang, 

Thank you for submitting your manuscript entitled "Phosphorylation of a tRNA synthetase by ATM/ATR essential for hypoxia-induced angiogenesis" for consideration as a Research Article by PLOS Biology.

Your manuscript has now been evaluated by the PLOS Biology editorial staff as well as by an academic editor with relevant expertise and I am writing to let you know that we would like to send your submission out for external peer review.

Please re-submit your manuscript within two working days, i.e. by Dec 03 2019 11:59PM.

Kind regards,

Ines

--

Ines Alvarez-Garcia, PhD

Senior Editor

PLOS Biology

Carlyle House, Carlyle Road

Cambridge, CB4 3DN

+44 1223–442810

---

## [Decision Letter · Decision Letter 1]

22 Jan 2020

Dear Xianglei,

Thank you very much for submitting your manuscript "Phosphorylation of a tRNA synthetase by ATM/ATR essential for hypoxia-induced angiogenesis" for consideration as a Research Article at PLOS Biology. Thank you also for your patience as we completed our editorial process, and please accept again my apologies for the delay in providing you with our decision. Your manuscript has been evaluated by the PLOS Biology editors, an Academic Editor with relevant expertise, and by three independent reviewers. Based on their specific comments and following discussion with the academic editor, I regret that we cannot accept the current version of the manuscript for publication. 

The reviews of your manuscript are appended below. As you will see, the reviewers find your results potentially interesting, however they also raise several concerns that would need to be addressed before we can consider your paper for publication. The reviewers feel the conclusions need to be strengthened with further experiments and also that the scope is a bit narrow and should be made broader. All the reviewers suggest several experiments to do so and ask for clarifications.

We remain interested in your study and we would be willing to consider resubmission of a comprehensively revised version that thoroughly addresses all the reviewers' comments. We cannot make any decision about publication until we have seen the revised manuscript and your response to the reviewers' comments. Your revised manuscript would be sent for further evaluation by the reviewers.

We appreciate that these requests represent a great deal of extra work, and we are willing to relax our standard revision time to allow you six months to revise your manuscript. We expect to receive your revised manuscript within 6 months.

**IMPORTANT - SUBMITTING YOUR REVISION**

*NOTE: In your point by point response to to the reviewers, please provide the full context of each review. Do not selectively quote paragraphs or sentences to reply to. The entire set of reviewer comments should be present in full and each specific point should be responded to individually, point by point.

*Resubmission Checklist*

*Published Peer Review*

*PLOS Data Policy*

*Blot and Gel Data Policy*

Sincerely,

Ines

--

Ines Alvarez-Garcia, PhD

Senior Editor

PLOS Biology

Carlyle House, Carlyle Road

Cambridge, CB4 3DN

+44 1223–442810

Reviewers’ comments

Rev. 1:

General Comment

In this manuscript, the authors proposed a mechanism leading to SerRS inactivation that enables cMyc- and HIF1a-mediated expression of VEGFA in response to hypoxia.

The authors showed that hypoxia activates ATM/ATR, which subsequently phosphorylate SerRS on specific residues (S101 and S241). Phosphorylation of SerRS decreases the its affinity for the VEGFA promoter. Thus, ATM/ATR release transcriptional repression that SerRS has on VEGFA promoter. This affects the ability of SerRS to induce EC proliferation and angiogenesis in vitro and in vivo.

The authors have previously shown that SerRS transcriptionally represses VEGFA by direct DNA binding to its promoter region and recruitment of the histone deacetylase Sirt2 (Ref 19). In addition, they had shown that SerRS counteracts the transcriptional activation activity of cMyc on VEGFA promoter. Thus, the major novelty in this work is the concept that ATM/ATR regulates SerRS as a transcriptional repressor in response to hypoxia. It is worth noting that a similar claim has been published recently by Song Y et al. Cancers (Basel) 2019 Nov 22;11(12). Song Y et al. reported the same residues being targeted by ATM/ATR kinases, and used similar approaches, including SerRS point mutants. Yet, their focus as a trigger for ATM/ATR activation was UV-mediated DNA double-strand breaks.

These findings are conceptually relevant, the manuscript is wells structured and experimental designs well organized.

For further clarity and confirmation of the proposed working model, authors should address the following major and minor issues:

Major points:

1. One of the major claims related to this work is that ATM/ATR are activated by hypoxia, which in turn negatively regulates SerRS by phosphorylation, leading to activation of VEGF expression. However, in this study, hypoxia was induced by a chemical approach. Glucose oxidase has been reported to induced H2O2, which induces DNA damage. Do authors' hypoxia protocol generates DNA damage? This would be a source of confounding effects related to the method of ATM/ATR activation. Please confirm with gammaH2AX stainings, or alternative methods. Can authors replicate the key results with a hypoxic chamber?

2. Authors should confirm that inhibition of ATM/ATR leads to reduced VEGF expression under hypoxic conditions (Fig.3H) using siRNA-mediated KD [single and combination] of ATM and ATR. Similarly, authors should confirm results in Fig.1I using the inhibitors.

3. In Fig.2A, it is not clear why overexpression of SerRS WT would lead to a decrease in VEGFA production in non-hypoxic conditions? Does that mean that SerRS is rate limiting? Yet, it still responds with the same magnitude to hypoxic stimulus Fig.3G. Can authors comment on this observation?

4. In Fig.3B/F, it is surprising that the binding of SerRS AA to VEGFA promoter is not greater than that of SerRS WT in normoxic but specially in hypoxic conditions. Could the authors comment on this? Since SerRS AA does prevent HIF1a and Myc from binding to VEGFA promoter (Fig.3F), this suggests that SerRS AA interacts with cMyc and HIF1a but not at the DNA level, thus, sequestering them and preventing their binding to VEGFA promoter. Changes in protein levels and/or post-translational modifications on HIF1a and Myc in response to hypoxia could also influence the output of the competition.

To clarify these different possibilities the authors could:

a. Perform an EMSA to evaluate a possible competition between SerRS WT and SerRS AA together with c-MYC for binding to VEGFA proximal promoter.

b. Perform a WB for HIF1a and c-Myc/c-Myc phosphorylated under the experimental conditions shown in Figure 3D and E.

c. Attempt to co-IP SerRS isoforms together with HIF1a or Myc.

d. Perform the same experiment as in Figure 3F but using the SerRS DD construct to show if absence of SerRS DD binding to DNA would make the cMyc and HIF1a binding as high as with SerRS WT version.

5. The interpretation of zebrafish experiments is a bit puzzling and should be further discussed in the text. The rescue of endogenous SerRS KD by both hSerRS WT and hSerRS AA tends to say that VEGFA is actively repressed in zebrafish embryos in cells that do not contribute for normal angiogenesis at these stages. Otherwise, it could be expected that hSerRS AA would abrogate VEGFA production and thus impair angiogenesis. Does this mean that ATR/ATM-mediated regulation of VEGFA production does not apply to zebrafish angiogenesis? Please comment in results/discussion.

Minor points

General

1. It is not clear from the figure legends or Material and Methods how many biological replicates were performed in WB experiments, for instance Fig.1F-I. If more than 1 experiment was performed, quantifications would be valuable.

Abstract

2. The following sentence does not accurately describe the results, since results show that hypoxia does not affect the binding efficiency of SerRS to the VEGFA promoter:

"Moreover, expression of SerRS S101A/S241A, a phosphorylation-deficient and constitutively active mutant, prevents hypoxia-induced DNA binding, which is necessary to activate c-Myc and HIF-1 for binding to the VEGFA promoter." Please correct.

Results

1. It is not clear from the figure legends or Material and Methods how many biological replicates were performed in WB experiments, for instance Fig.1F-I. If more than 1 experiment was performed, quantifications would be valuable.

2. For clarity, the authors should describe in the text what "substrate (p-S*Q) antibody" is.

3. For clarity, the authors should improve the description of the results in Figure 1E. Specifically, describe (in the text or figure legend) what Ni-NTA is and describe in the text the result for pATM, pRPA.

4. The title of the section "Phosphorylation inactivates SerRS as a transcriptional repressor in cells and in vivo" does not really match the findings described in this same section. In this section, the authors show that SerRS phosphorylation affects VEGFA expression and affects angiogenesis in vivo but do not show that phosphorylation affects SerRS transcriptional repressor activity. The effect of phosphorylation on transcriptional repressor activity of SerRS is shown only in the next section. Therefore, the title of the section should be rephrased to match more exactly the findings.

5. In Fig.6, it is very surprising that no tumor growth curves were measured.

6. In Fig.4C/D, data on anti-VEGF antibody. Do the results mean that VEGF has no role in basal conditions? Please comment.

7. Experiments using matrigel in Fig.4. The description in the text and figure legend is very unclear for readers. It is not obvious which cells were transfected and/or subjected to hypoxia or anti-VEGFA. Authors should make label figure in a more precise way so it becomes obvious that HUVECs in matrigel were not subjected to hypoxia and transfection, but rather only HUVECs that produced the supernatant.

8. Figure 5 B, 6A and 6C - stainings are not easy to see - consider changing colors to facilitate visualization (e.g. - grayscale inverted + green). In addition, the resolution of the images is very low. In addition, the resolution of input images and file compression should be revised.

Discussion

1. The Figure S7 lacks details (efficiency of SerRS knockdown, how the experiment was performed (timing of analysis after shRNA transfection in the figure legend). Also, the hypothesis that SerRS transcriptionally regulates IL8 expression (as a direct target) and that it can antagonize ROS/NF-kB/IL8 hypoxia response is, at this stage, highly speculative. Therefore, given the lack of solid evidence for such conclusion, I suggest the authors the removal this part from the working model in Figure 7.

Figure legends

1. Figure 1E - Ni-NTA should be described in the figure legend.

2. Figure 3B - In the figure, fold change is shown. In the legend, it should be specified how the fold change is represented.

Materials and Methods

1. It would be helpful to provide more accurate information on the following: "Hypoxia medium was prepared by diluting glucose oxidase and catalase at a constant 1:10 ratio in cell culture medium". Could you please specify the final concentration in solution of these enzymes?

2. Anti-HIF-1β (ARNT, #3414) antibody was not used in the study. It should be removed from the the Materials and methods section.

Typographical errors

The MS contains several typos and imprecisions that should be corrected. Just to highlight a very obvious one, the title "Phosphorylation of a tRNA synthetase by ATM/ATR essential for hypoxia-induced angiogenesis" is missing a verb.

A few other below:

1. All Western Blot figures - "KD" refers to kilodaltons, should be spelled "kDa"

2. Figure 1 B - y axis - "VEGFA indiction" should be spelled "VEGFA induction"

3. Figure 1 C - "H. sapiesn" should be spelled "H. sapiens"

4. Figure 1 E - "Nulear extracts" should be spelled "Nuclear extracts"

5. Abstract: "Moreover, expression of SerRSS101A/S241, a phosphorylation-deficient and constitutively active mutant…" should be spelled "Moreover, expression of SerRSS101A/S241A, a phosphorylation-deficient and constitutively active mutant…"

6. Results: "We further confirmed the expression of SerRS and the molecular markers for vasculature in the tumours xenografts by Western bolt analysis" should be spelled "We further confirmed the expression of SerRS and the molecular markers for vasculature in the tumours xenografts by Western blot analysis"

7. Discussion: "Tumours cells can adapt to the hypoxic environment through orchestrating the expression of a large number of genes that functions in apoptosis, cell survival, energy metabolism, and angiogenesis" should be spelled "Tumour cells can adapt to the hypoxic environment through orchestrating the expression of a large number of genes that function in apoptosis, cell survival, energy metabolism, and angiogenesis"

8. Discussion: "or example, K-RAS and hypoxia tension can synergistically increase the level of GTP-bound Rho" should be spelled: "For example, K-RAS and hypoxia tension can synergistically increase the level of GTP-bound Rho"

Rev. 2:

In this submitted manuscript, Shi et al. performed a series of studies in human cells, zebrafish, and mice, to identify a novel molecular mechanism in hypoxia-induced VEGF in both endothelial cells and cancer cell line. Hypoxia phosphorylates SerRS at Ser101 and Ser241 through ATM/ATR, which diminishes the binding of SerRS in the promoter region of VEGF and attenuates the transcriptional repress effect of SerRS. Furthermore, they found the dissociation of SerRS in the promoter of VEGF is required for HIF1a and c-Myc binding. As the functional readout, SerRS WT or SerRS AA abolishes hypoxia-induced angiogenesis. Overall, the study is nicely executed and the conclusion is convincing. Addressing the following points could further strengthen the manuscript.

1. The involvement of SerRS phosphorylation, HIF1a, c-Myc, and VEGF expression under hypoxia has been well documented. The regulation between SerRS phosphorylation and HIF1a, c-Myc binding is interesting. The mechanistic study could be further strengthened by addressing the relationship and balance between transcriptional repressors and activators in the VEGF promoter. For example, why SerRS dissociation from the VEGF promoter is required for HIF1a and c-Myc binding? Whether the SerRS binding site is near to the HIF1a or c-Myc binding site? The authors showed the SerRS AA form abolished hypoxia-induced c-Myc and HIF1a binding. To further validate this finding, the author should study whether deletion or mutation of SerRS binding site in a reporter system (e.g., VEGF promoter-Luci reporter plasmid) can rescue the SerRS AA decreased c-Myc or HIF-1a binding and gene expression?

2. In HUVECs, is the hypoxia-induced phosphorylation of SerRS also through ATM/ATR? The authors should consider to detect p-ATM in HUVECs under hypoxia (Figure 1G).

3. In Figure 2B, under SerRS knockdown (SerRS-MO), SerRS AA caused an increase in VEGF mRNA expression when compared with SerRS WT. As a de-phosphomimetics form, SerRS AA should cause more transcriptional repression effect than Ser WT, at least should have the comparable VEGF expression level under normoxia condition. The authors need to clarify this discrepancy.

4. In Figure 4A and 4B, the comparisons were made between SerRS WT and SerRS AA under normoxia and hypoxia. Since SerRS DD strongly induced SerRS release from VEGF promoter, a comparison between SerRS WT and SerRS DD is essential to reveal the VEGF regulatory effect of SerRS phosphorylations.

5. Data showed in Figure 1 and Figure 3 showed that SerRS had a low level of phosphorylation and high level of DNA binding under normoxia. If so, SerRS WT and SerRS AA should have comparable DNA binding. However, Figure 3F shows that the DNA binding of SerRS was decreased in SerRS AA transfection when compared with WT. SerRS DD form should be involved in Figure 3F, especially under normoxia condition.

6. As the major evidence of endothelium related angiogenesis, the CD31 or vWF staining is needed in the figure 5.

7. The last subtitle in Results “SerRS AA strongly inhibits tumours angiogenesis in mice”, but the data indicated there was no extra suppression effect of SerRS AA in VEGF expression and angiogenesis when compared with WT. As discussed by the authors, this might be due to the high level of SerRS WT overexpression saturated the phosphorylation capacity of ATM/ATR. But in vitro data showed the decreased VEGF and angiogenesis in AA compared with WT (e.g., Figure 4), even under the much higher level of SerRS overexpression. The authors should future discuss the discrepancy or weaken this subtitle.

8. The western blot in Figure 6E showed the higher expression levels of SerRS in the AA group than the WT group, but the bar graph in Figure 6F presented the opposite trend.

9. The detailed methods about SerRS WT, SerRS AA, or SerRS DD stable transfection in HUVECs are needed.

10. In Figure 1B, the labeling on Y-axis should be “VEGFA induction”.

Rev. 3:

In this study the authors analyze the mechanism of hypoxia induced VEGFA expression, by elucidating the transcriptional repressor role of SerRS in normoxic conditions. By combining analysis in human cells, zebrafish and mice, they show that SerRS phosphorylation through ATM and ATR is induced by hypoxia and reduces SerRS DNA binding.

The study is generally well done, experimentally sound and through the comparison of the different models does validate most of the claims by more than one way. Therefore it does not require much additional experimental work for publishing. It is original enough and of importance to researchers in its field or at least to a subset of this audience However, I doubt it to be of interest to a more general audience, as would be required for publication in Plos biology in its current state. To broaden the impact, the authors should analyze the consequences of SerRS phosphorylation for other c-Myc and Hif-1 targets, ideally even establishing differences between hypoxia dependent and independent targets (at least of c-Myc).

Minor comments:

The manuscript is well organized and written clearly to allow for understanding of non-specialist. However, throughout the manuscript there is a large number of grammatical and spelling errors, which need to be corrected.

Also there seems to be insecurity in terms of proper nomenclature. As the nomenclature rules are different for mouse, human and fish, the authors do need to check each gene/protein and abbreviation for species correct nomenclature.

The last sentence of the abstract is pure speculation and meaningless, especially dealing with intracellular transcription factors, which make poor therapeutical targets, therefore the authors should focus on a more strong and study related finish of the abstract.

---

## [Decision Letter · Decision Letter 2]

16 Oct 2020

Dear Dr Yang,

Thank you for submitting your revised Research Article entitled "Phosphorylation of a tRNA synthetase by ATM/ATR is essential for hypoxia-induced angiogenesis" for publication in PLOS Biology. Please accept again my apologies for the delay in the process. I have now obtained advice from two of the original reviewers and have discussed their comments with the Academic Editor. 

Based on the reviews (attached below), we will probably accept this manuscript for publication, assuming that you will modify the manuscript to address the remaining points raised by Reviewer 1. In addition, we would like you to consider a change in the title to make it more precise to: "Phosphorylation of seryl-tRNA synthetase by ATM/ATR is essential for hypoxia-induced angiogenesis"

Please also make sure to address the data and other policy-related requests noted at the end of this email.

We expect to receive your revised manuscript within two weeks. Your revisions should address the specific points made by each reviewer. In addition to the remaining revisions and before we will be able to formally accept your manuscript and consider it "in press", we also need to ensure that your article conforms to our guidelines. A member of our team will be in touch shortly with a set of requests. As we can't proceed until these requirements are met, your swift response will help prevent delays to publication.

- a cover letter that should detail your responses to any editorial requests, if applicable

*Copyediting*

*Published Peer Review History*

*Early Version*

Sincerely,

Ines

--

Ines Alvarez-Garcia, PhD,

Senior Editor,

ialvarez-garcia@plos.org,

PLOS Biology

DATA POLICY:

Fig. 1A, B; Fig. 2A, B; Fig. 3B, D, E, F, G, H, I; Fig. 4A, B, D; Fig .5C; Fig. 6A, C, E, F, H, J, L; Fig. S4; Fig. S6B; Fig. S7B; Fig. S8C; Fig. S9A, B and Fig. S10A, C

Reviewers’ comments

Rev. 1:

The reviewer acknowledges the substantial efforts that were made by the authors to answer all queries. These efforts are even more commendable given the ongoing pandemic restrictions. Authors have answer most of the major and minor concerns and they have clarified many of the initial critics. Therefore, the reviewer supports the publication of this report in PloS Biology. However, there are still some concerns and improvements in the current MS format (related to the first review and not requiring experimental work) that need to be addressed before publication.

Major Point 1- Authors should provide quantification of WB for new Fig.S2.

Major Point 4- new Fig.3F and old Fig.3F differ significantly in the data for SerRS-AA. When looking at the error bars, one would guess that authors exclude old data points. Does this graph only show data for the new experiment? What is the rational to exclude the previous (old) data?

Major Point 5- authors should provide a reference for the following statement "…because the zebrafish are not under hypoxia;".

Moreover, as authors said, zebrafish are not under hypoxia, and thus, ATM/ATR will not be active to phosphorylate SerRS. From these experiments the major conclusion is that SerRS is a constitutive repressor of VEGF expression in zebrafish embryos, unrelated to ATM/ATR activity, as published in https://elifesciences.org/articles/02349.

Overall, the zebrafish data does not bring major novelties to the biology of ATM/ATR regulation of SerRS, and the connection between Fig.1 and Fig.2 is non-existent. Thus, zebrafish data should be included as supplementary data. Moreover, the conclusion that ATM/ATR is not involved in SerRS regulation in zebrafish should be made even clearer in the MS.

Minor Point 1- The authors have addressed the major concern on the inaccuracy of the sentence. Yet, the new sentence, "Moreover, expression of SerRSS101A/S241A, a phosphorylation-deficient and constitutively active mutant, prevents hypoxia-induced c-Myc and HIF-1 binding to the VEGFA promoter, which is necessary for the transcription factors to activate gene expression.", is still somehow difficult to read.

The reviewer suggests the following:

"Moreover, expression of SerRSS101A/S241A, a phosphorylation-deficient and constitutively active mutant, prevents the hypoxia-mediated binding of c-Myc and HIF-1 to the VEGFA promoter, two transcription factors that activate VEGFA expression."

Minor Point 5 - These new experiments are puzzling. Authors should clarify why SerRS WT overexpression leads to a significant decrease in tumour growth and VEGFA production in tumors. This is incompatible with the model proposed and with experiments showing that overexpression of SerRS WT does not affect levels of VEGFA expression in hypoxic conditions relative to control cells (Fig3G, Fig4A/B). As authors wrote in reply to major point 3 "The WT SerRS (whether it is overexpressed or endogenous) can be inactivated by phosphorylation under hypoxia conditions. This is why cells expressing only endogenous SerRS (Vector control) and cells overexpressing SerRSWT can respond to a similar magnitude to hypoxic stimulus (in term of VEGFA induction) as shown in Fig. 3G." Authors should clarify the discrepancy.

Minor Point 7- Authors did not interpret correctly the reviewer statement.

Fig.4 does use Matrigel for the tube formation assay, and the experiment uses HUVECs, as described in figure legend and material and methods. Please re-interpret the original comment and adapt accordingly. Authors should make clearer that only conditional media, from the defined conditions, was used, instead of using different cell lines assay with normoxia and hypoxia conditions directly on the tube formation assay. The reviewer suggests that a cartoon describing the experimental setup could guide the readers for this important difference.

Rev. 2:

The authors have done an outstanding revision and the manuscript has been significantly strengthened. Since the authors addressed all my concerns in the revised manuscript, I suggest an acceptance of the manuscript.

---

## [Editor Report · Decision Letter 3]

19 Nov 2020

Dear Dr Yang,

On behalf of my colleagues and the Academic Editor, Tanya Paull, I am pleased to inform you that we will be delighted to publish your Research Article in PLOS Biology. 

PRODUCTION PROCESS

Before publication you will see the copyedited word document (within 5 business days) and a PDF proof shortly after that. The copyeditor will be in touch shortly before sending you the copyedited Word document. We will make some revisions at copyediting stage to conform to our general style, and for clarification. When you receive this version you should check and revise it very carefully, including figures, tables, references, and supporting information, because corrections at the next stage (proofs) will be strictly limited to (1) errors in author names or affiliations, (2) errors of scientific fact that would cause misunderstandings to readers, and (3) printer's (introduced) errors. Please return the copyedited file within 2 business days in order to ensure timely delivery of the PDF proof. 

If you are likely to be away when either this document or the proof is sent, please ensure we have contact information of a second person, as we will need you to respond quickly at each point. Given the disruptions resulting from the ongoing COVID-19 pandemic, there may be delays in the production process. We apologise in advance for any inconvenience caused and will do our best to minimize impact as far as possible.

EARLY VERSION

PRESS 

Kind regards,

Erin O'Loughlin

Publishing Editor, 

PLOS Biology

on behalf of

Ines Alvarez-Garcia,

Senior Editor

PLOS Biology